# MORA: Improving Ensemble Robustness Evaluation with Model-Reweighing Attack

**Yunrui Yu***
State Key Lab of IOTSC,
University of Macau
Macau SAR, China
`yb97445@um.edu.mo`

**Xitong Gao***
Shenzhen Institute of Advanced Technology,
Chinese Academy of Sciences
Shenzhen, China
`xt.gao@siat.ac.cn`

**Cheng-Zhong Xu**[†]
State Key Lab of IOTSC,
University of Macau
Macau SAR, China
`czxu@um.edu.mo`

## Abstract

Adversarial attacks can deceive neural networks by adding tiny perturbations to their input data. Ensemble defenses, which are trained to minimize attack transferability among sub-models, offer a promising research direction to improve robustness against such attacks while maintaining a high accuracy on natural inputs. We discover, however, that recent state-of-the-art (SOTA) adversarial attack strategies cannot reliably evaluate ensemble defenses, sizeably overestimating their robustness. This paper identifies the two factors that contribute to this behavior. First, these defenses form ensembles that are notably difficult for existing gradient-based method to attack, due to gradient obfuscation. Second, ensemble defenses diversify sub-model gradients, presenting a challenge to defeat all sub-models simultaneously, simply summing their contributions may counteract the overall attack objective; yet, we observe that ensemble may still be fooled despite most sub-models being correct. We therefore introduce MORA, a model-reweighing attack to steer adversarial example synthesis by reweighing the importance of sub-model gradients. MORA finds that recent ensemble defenses all exhibit varying degrees of overestimated robustness. Comparing it against recent SOTA white-box attacks, it can converge orders of magnitude faster while achieving higher attack success rates across all ensemble models examined with three different ensemble modes (*i.e.*, ensembling by either softmax, voting or logits). In particular, most ensemble defenses exhibit near or exactly $0\%$ robustness against MORA with $\ell^\infty$ perturbation within 0.02 on CIFAR-10, and 0.01 on CIFAR-100. We make MORA open source with reproducible results and pre-trained models; and provide a leaderboard of ensemble defenses under various attack strategies[1].

## 1 Introduction

Many safety-critical applications, such as autonomous robots [34], self-driving [8], search engines [24], *etc.* are becoming increasingly powerful and reliant on deep neural networks (DNNs). Despite the monumental success of DNNs on these applications, they are highly susceptible to adversarial examples — an attacker can add tiny delibrate perturbations to the input data, misleading the model into giving incorrect results [23, 9]. Such adversarial attacks could pose a significant threat to the safety and reliability of deep learning applications.

To mitigate this threat, many defense strategies [17, 33, 4] based on adversarial training [17] have been proposed to improve model robustness. Adversarial training, however, gains robustness at the

---

*Xitong Gao and Yunrui Yu contributed equally to this work.

[†]Correspondence to: Cheng-Zhong Xu (`cz.xu@um.edu.mo`).

[1]`https://github.com/lafeat/mora`

36th Conference on Neural Information Processing Systems (NeurIPS 2022).

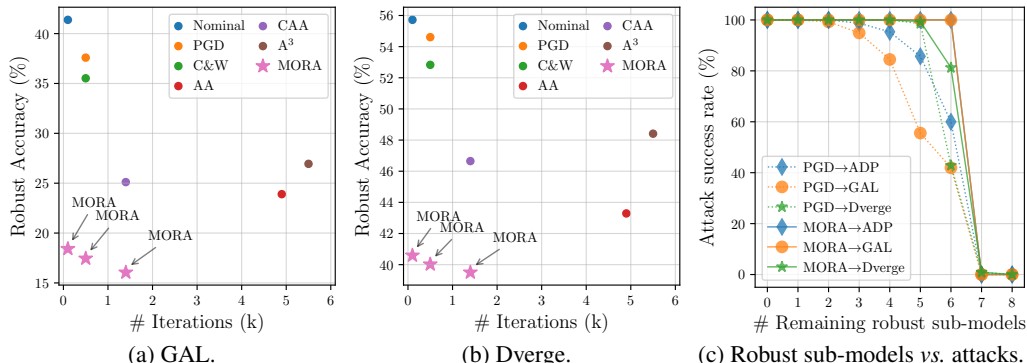

Figure 1: (a,b) Existing attacks [17, 3, 5, 18, 16] with strong baselines are neither efficient in the number of model forward/backward passes, nor reliable in the estimation of ensemble robustness when compared with MORA. GAL [14] and Dverge [30] defenses are trained on CIFAR-10 with 8 sub-models. We used $\ell^\infty$ attacks within $\epsilon = 0.01$, "Nominal" is self-reported. (c) MORA can successfully fool logit-based ensembles (ADP [19], GAL and Dverge) even with the majority of their sub-models giving correct outputs ("$A \to B$" means using $A$ to attack $B$ for up to 100 iterations).

expense of model accuracy on clean natural images [27]. Ensemble defenses [19, 14, 30, 31] have thus emerged to combine multiple predictions from independent sub-models. The intuition is that an ensemble of models can often lead to higher accuracy, while learning to stop adversarial example transfer among sub-models may improve robustness against adversarial attacks. This approach could potentially offer a promising research direction to improve model robustness while preserving high accuracy on natural inputs.

Yet surprisingly, under the white-box threat model, existing state-of-the-art (SOTA) adversarial attacks with strong performance on conventional DNN models performed poorly on ensemble models, sizeably overestimating their robustness (Figures 1a and 1b). This also suggests, to some extent, that ensemble defenses may rely on two potential design flaws below that cause obfuscated gradients [1], *i.e.*, they are either deliberately non-differentiable, or give no useful gradients, thus inducing overestimated robustness:

(a) *Gradient obfuscation via ensemble-forming strategy.* They typically form ensembles by averaging probability vectors (softmax) of sub-models, and softmax operations can easily cause gradient obfuscation. While the model's actual robustness is pertinent to the strategy used to form an ensemble, this indicates that gradient-based attacks have to *also* leverage this effectively.

(b) *Gradient diversification.* Motivated by the reasoning that a majority of sub-models may need to be fooled for successful attacks, they learn to reduce adversarial transferability among sub-models, often via gradient diversification. This intuitively causes sub-models to counteract each other, averaging to a small or inaccurate overall gradient. Attacking only the ensemble loss would fool most sub-models, but the ensemble may remain still correct; conversely, it is actually possible to fool an ensemble, despite the majority of its sub-models giving correct predictions (Figure 1c).

From the above observations, it is perceivable that the practical evaluation of ensemble robustness cannot be solely done by treating such models holistically. To this end, this paper introduces MORA, model-reweighing attack, to adaptively adjust the importance of sub-models in attack iterations. Sub-models are reweighed according to their respective "ease of attack", which is in turn evaluated by the gradient of the difference of ensemble classification outputs *w.r.t.* the ones of individual sub-models. Pushing the limits of the current SOTA in ensemble robustness evaluation, it draws inspiration from recent effective attack tactics, *e.g.*, momentum [7, 5], step size schedule [5, 16], loss normalization [32], and multiple targets [5, 25]. We summarize our contributions:

- This paper presents the first extensive study on the robustness of ensemble defenses under multiple ensemble-forming strategies.
- By reweighing the importance weights of sub-models to steer adversarial example synthesis, we show that gradient-based attacks on ensemble defenses can often be orders of magnitude faster, while enjoying a higher success rate.

- Empirical results on a wide variety of different ensemble defenses show that MORA out-performs competing attacks in both performance and convergence rate. Finally, this paper provides extensive ablation of its components and sensitivity analyses of hyperparameters.

To our best knowledge, MORA is currently the strongest attack against a wide range of ensemble defenses. We make MORA open source with reproducible results and pre-trained models; moreover, we maintain a leaderboard of ensemble defenses under various attack strategies.

## 2 Preliminaries & Related Work

### 2.1 Adversarial Attacks

An adversarial example adds a small perturbation, typically bounded a small value with $\ell^p$ norms, to the original image, such that the model under attack can be deceived into giving incorrect results. The advent of adversarial attacks [23] has piqued the interest of deep learning practitioners, and revealed security concerns of deep learning [26, 21], improved GAN training [2] transfer learning [28, 6], and DNN interpretability [22], *etc.* Formally, assuming a defending classifier $f : \mathcal{I} \to \mathbb{R}^K$, taking an input image $\mathbf{x} \in \mathcal{I} = [0,1]^{C \times H \times W}$ as input, and $\arg\max f(\mathbf{x})$ predicts the correct class label $y \in \mathcal{C}$, then an attacker attempts to find an adversarial example $\hat{\mathbf{x}}$ in the set:

$$\{\hat{\mathbf{x}} \in \mathcal{A}_{\epsilon,\mathbf{x}} : \ \arg\max f(\hat{\mathbf{x}}) \neq y\}. \tag{1}$$

Here, $\hat{\mathbf{x}} \in \mathcal{A}_{\epsilon,\mathbf{x}}$ constrains the adversarial example $\hat{\mathbf{x}}$ to be within both the input space $\mathcal{I}$ and a small $\epsilon$-ball of $\ell^p$-distance from the original image $\mathbf{x}$, or equivalently $\|\mathbf{x} - \hat{\mathbf{x}}\|_p \leq \epsilon$. Satisfying the condition $\arg\max f(\mathbf{x}) \neq y$ means that $f(\mathbf{x})$ fails to give the expected correct classification $y$. We focus on the $\ell^\infty$ white-box threat model commonly considered by the defenses examined in this paper, which grants the attacker completely access to the internals of the defender, including, for instance, its model architecture, parameters, training algorithms, *etc.*

One of the popular and effective white-box attacks used by many defenders to evaluate their robustness is *projected gradient descent* (PGD) [17], which finds adversarial examples by maximizing the classification loss with gradient descent:

$$\hat{\mathbf{x}}_{i+1} = \mathcal{P}_{\epsilon,\mathbf{x}}(\hat{\mathbf{x}}_i + \alpha_i \operatorname{sign}(\nabla \mathcal{L}(f(\hat{\mathbf{x}}_i), y))), \tag{2}$$

where $\mathcal{L}$ is typically the softmax cross-entropy (SCE) loss used to train the model, $\alpha_i$ is the step size, and we let the initial $\hat{\mathbf{x}}_0 \triangleq \mathcal{P}_{\epsilon,\mathbf{x}}(\mathbf{x} + \boldsymbol{\mu})$. The projection function $\mathcal{P}_{\epsilon,\mathbf{x}}(\mathbf{v})$ constrains its input $\mathbf{v}$ to be within the feasible region $\mathcal{A}_{\epsilon,\mathbf{x}}$, and finally $\boldsymbol{\mu} \sim \mathcal{U}(-\epsilon, \epsilon)$ is a uniformly distributed noise bounded by $[-\epsilon, \epsilon]$. Besides PGD, C&W [3] is also a gradient-based attack which, instead of projection, indirectly constrains the search space by regularization.

As PGD gains popularity, many defense mechanisms rely on it to evaluate their robustness. Unfortunately, AutoAttack (AA) [5] finds that many of the defenses may inadvertently break PGD-based attacks, which result in drastic overestimation of their robustness, and proposes to combine an ensemble of diverse attacks to minimize robustness overestimation. LAFEAT [32] learns to leverage intermediate layers of the DNN, and shows that attacking multiple layers can produce stronger attacks, but unfortunately it cannot be applied to ensemble defenses. Adaptive Auto Attack ($A^3$) [16] improves attack success rates by using the gradient directions to prescribe a more effective initial random perturbation. As defenders may design mechanisms to circumvent existing attacks, Adaptive attacks [25] manually tailor specific attack strategies for an extensive set of defenses. Finally, Composite Adversarial Attacks (CAA) [18] further combine a large selection of attack methods, and use a genetic algorithm to learn an optimal attacking sequence. In comparison, MORA is a unified approach which uses only one attack algorithm, does not require a compute-intensive learning procedure, and yet it still achieves fast and SOTA estimation of ensemble robustness.

### 2.2 Defending Against Adversarial Attacks

Defending against adversarial attacks can be defined as a saddle-point problem to minimize the training loss on adversarial examples [17] with samples $(\mathbf{x}, y)$ drawn from the training set:

$$\min_{\boldsymbol{\theta}} \mathsf{E}_{(\mathbf{x},y)} \left[ \max_{\hat{\mathbf{x}} \in \mathcal{A}_{\epsilon,\mathbf{x}}} \mathcal{L}(f(\hat{\mathbf{x}}), y) \right], \tag{3}$$

where $\mathcal{L}$ is the training loss, *e.g.*, the SCE loss. A direct optimization-based approach to approximately solving the above problem is *adversarial training* [17], *i.e.*, to train the DNN model with its own adversarial examples. Training DNNs to be robust is, however, a challenging endeavor. First, it may be much more compute intensive as training examples are typically generated with PGD [17], requiring a few forward/backward passes of the DNN. Second, to avoid overfitting, it requires stopping training early, a much larger size of the training set [4], and using improved data augmentation [20] or generated data [10]. Thirdly, as noted by other literatures [5, 32], currently no other design choices can rival the robustness provided by adversarial training, and notably, many defense strategies are considered harmful to model robustness [25]. Finally, the resulting models often cannot achieve high clean accuracy [27].

### 2.3 Ensemble Defenses & Ensemble-forming Strategies

Ensemble-based defense techniques may pave an alternative path to address the challenges of adversarial robustness, as they could potentially work around the above limitations of adversarial training. Adopting the theme of minimizing adversarial example transferability across sub-models, each ensemble defense proposed unique solutions. ADP [19] increases the orthogonality of non-maximal class logits among sub-models to encourage diversity. GAL [14] minimizes a gradient alignment loss, which directly reduces the cosine-similarity between sub-models. Building on top of this, TRS [31] further regularizes the smoothness of the loss function, as gradient orthogonality with smoothness may further diversify sub-models. Dverge [30] instead uses the adversarial examples of a sub-model to train another sub-model, thus lowering transferability. Ensemble defenses are also particularly interesting, as they are the last line of defense against even the strongest existing white-box attacks without resorting to adversarial training, showing a certain degree of robustness.

Besides the above mechanisms for training a successful ensemble defense, there exists different ways to combine sub-model predictions. Let us assume that an ensemble defense trains $M$ sub-models, $f_m : \mathcal{I} \rightarrow \mathbb{R}^K$ for $m \in [1 : M]$, an ensemble $f_E : \mathcal{I} \rightarrow \mathbb{R}^K$ thus forms a final classification result by combining individual decisions from the sub-models, namely:

$$f_{\mathrm{E}}(\mathbf{x}) = \tfrac{1}{M} \sum_{m \in [1:M]} \mathrm{ens}(f_m(\mathbf{x})), \tag{4}$$

where $\mathrm{ens}$ is the ensemble-forming operator. In this paper, we investigate $\mathrm{ens} \in \{\mathrm{softmax}, \mathrm{wta}, \mathrm{id}\}$, where the potential choices respectively denoting forming an ensemble from sub-model outputs $f_m(\mathbf{x})$ by either summing predicted probabilities (evaluated with $\mathrm{softmax}$), or majority votes (using $\mathrm{wta}$, the winner-take-all operator), or simply summing the logits (with $\mathrm{id}$, the identity operator). Defending ensemble methods [19, 14, 30, 31] tested in this paper all employed the $\mathrm{softmax}$-based strategy to report their robustness. Methods that are exceptions to these options exist, for instance, ECOC [29] allows sub-models to produce binary predictions, and use error correcting codes based on the Hamming distance to combine the predictions into classification outputs. This approach is unfortunately not robust, and the added complexity is error-prone and may harm robustness [25].

Moreover, as the voting ($\mathrm{wta}$) strategy is non-differentiable, an attacker can soften it approximately using a softmax operation with temperature $\tau$, where we used $\tau = 0.1$ universally:

$$\mathrm{softwta}_\tau(\mathbf{x}) \triangleq \mathrm{softmax}(\mathbf{x}/\tau). \tag{5}$$

## 3 The Model-Reweighing Attack (MORA)

### 3.1 Problem Formulation & High-Level Overview

As discussed in Section 1, existing ensemble defenses may obfuscate gradients with the ensemble-forming mode and gradient diversification, such that the final loss of the ensemble model can no longer provide effective signals for gradient descent. It is therefore desirable to find an alternative $\mathcal{L}$ to the original SCE loss $\mathcal{L}^{\mathrm{sce}}$ on the ensemble, such that for a given number of iterations $I$, the original $\mathcal{L}^{\mathrm{sce}}$ loss can be maximized:

$$\begin{aligned}
\max_{\mathcal{L}} \ \mathcal{L}^{\mathrm{sce}}(\hat{\mathbf{x}}_I, y) \quad \text{where} \quad &\hat{\mathbf{x}}_0 = \mathcal{P}_{\epsilon, \mathbf{x}}(\mathbf{x} + \boldsymbol{\mu}), \\
&\hat{\mathbf{x}}_{i+1} = \mathrm{PGD}(\mathcal{L}(f_E(\hat{\mathbf{x}}_i), f_{[1:M]}(\hat{\mathbf{x}}_i), y)),
\end{aligned} \tag{6}$$

and $\mathrm{PGD}(\cdot)$ denotes a PGD step along the gradient of loss function $\mathcal{L}(f_{\mathrm{E}}(\hat{\mathbf{x}}_i), f_{[1:M]}(\hat{\mathbf{x}}_i), y)$, which not only takes the ensemble predictions $f_E(\hat{\mathbf{x}}_i)$ as input, but can further utilize sub-model predictions

$f_{[1:M]}$ to guide the PGD iterations. The challenge at hand is, therefore, to find a suitable $\mathcal{L}$ which can generate attacks on ensemble defenses efficiently and effectively.

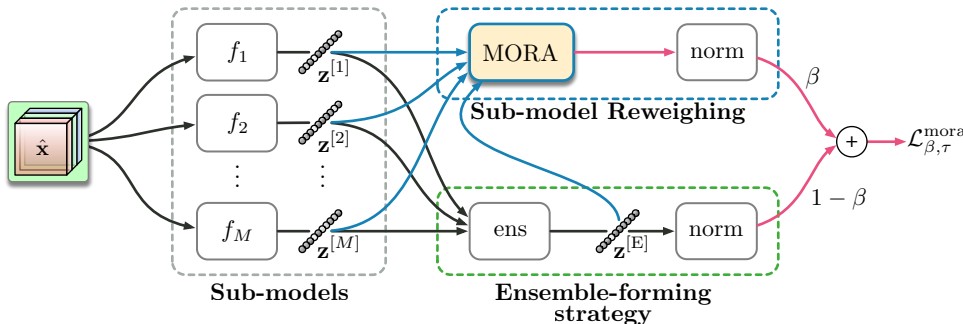

Figure 2: A high-level overview of MORA. Existing attack methods focus on maximizing the $\mathcal{L}^{\text{sce}}$ of the ensemble predictions $\mathbf{z}^{[\text{E}]}$. MORA further introduces a model-reweighing mechanism, which takes as input sub-models predictions $\mathbf{z}^{[1:M]}$, and the ensemble output $\mathbf{z}^{[\text{E}]}$, forming a combined loss $\mathcal{L}^{\text{mora}}_{\beta,\tau}$. MORA generates adversarial examples by maximizing the resulting loss.

MORA aims to provide a potential optimization route towards the above problem formulation. Namely, in addition to the original output of the ensemble $f_\text{E}(\hat{\mathbf{x}})$, we leverage the sub-model predictions $f_{[1:M]}(\hat{\mathbf{x}})$ to facilitate the optimization. By way of illustration, Figure 2 shows a high-level overview of the model-reweighing attack, where we compliment the ensemble loss, with a newly added sub-model reweighing loss $\mathcal{L}^{\text{mora}}_{\beta,\tau}$, as an auxiliary attack vector alongside the original objective. Not only can the new loss bypass the ensemble-forming strategy to work around its obfuscated gradients, but it further exploits information present in the individual sub-model and ensemble predictions to steer the direction of adversarial example synthesis.

## 3.2 Adaptive Sub-model Importance

Before we begin, assume that $\mathbf{z}^{[m]} \triangleq f_m(\mathbf{x})$ represents the $m^{\text{th}}$ sub-model output, and let $\mathbf{z}^{[m]}_t$ denote the corresponding logit of label $t$. We define the difference of logits (DL) [3] $k^{[m]} \triangleq \mathbf{z}^{[m]}_y - \mathbf{z}^{[m]}_{\hat{y}}$, which is the difference between the predictions of the ground truth $\mathbf{z}^{[m]}_y$ and the maximum of the remaining classes $\mathbf{z}^{[m]}_{\hat{y}} \triangleq \max_{i \in \mathcal{C}/y} \mathbf{z}^{[m]}_i$, where $\mathcal{C}/y$ is the set of all class labels except $y$. Similarly, we let $\mathbf{z}^{[\text{E}]} \triangleq f_\text{E}(\mathbf{x})$, and $\mathbf{z}^{[\text{E}]}_t$ and $k^{[\text{E}]}$ be the respective variants of the ensemble prediction. It is notable that a successful attack happens when $k^{[\text{E}]} < 0$, and similarly $k^{[m]} < 0$ means the $m^{\text{th}}$ sub-model is producing incorrect classification.

Ensemble defenses tend to diversify sub-model gradients, for instance, ADP [19] minimizes the cosine-similarity $\langle \nabla\ell(\mathbf{z}^{[a]}), \nabla\ell(\mathbf{z}^{[b]}) \rangle$ among each loss function gradient pairs of sub-models $\ell(\mathbf{z}^{[a]})$ and $\ell(\mathbf{z}^{[b]})$. Their intuition is that it may lower transferability among these sub-models, such that attacks with the overall gradient of the ensemble, $\textit{i.e.}$, $\nabla\ell(\mathbf{z}^{[\text{E}]}) = \frac{1}{M}\sum_{m \in [1:M]} \nabla\ell(\mathbf{z}^{[m]})$, are becoming less effective in misleading all sub-models simultaneously, as individual gradients in $\nabla\ell(\mathbf{z}^{[m]})$ are encouraged to be orthogonal to each other. To this end, we propose to reweigh the importance of sub-models, by instead considering the modified gradient:

$$\widehat{\nabla\ell(\mathbf{z}^{[\text{E}]})} = \frac{1}{M}\sum_{m \in [1:M]} \lambda^{[m]}_\tau(\mathbf{z}^{[m]}) \nabla\ell(\mathbf{z}^{[m]}), \tag{7}$$

where $\lambda^{[m]}_\tau$ assigns weights to important sub-models to contribute more heavily to the attack gradient.

While the adversarial examples of the ensemble could present a challenge to discover, individual sub-models are weak defenders which can be easily defeated. Based on this property, we propose to weigh sub-models importance with the rate of change in $k^{[\text{E}]}$ $\textit{w.r.t.}$ that of $k^{[m]}$, $\textit{i.e.}$, sub-models would be given higher weights if attacking it would bring a significant change to the ensemble's prediction. Following this idea, for all ensemble-forming strategies (softmax, voting, logits), we rewrite $k^{[\text{E}]}$ as a function of $k^{[m]}$, where the term below can become a function of $k^{[m]}$:

$$k^{[\text{E}]} = \text{ens}(\mathbf{z}^{[m]})_y - \text{ens}(\mathbf{z}^{[m]})_{\hat{y}} = \text{ens}(\mathbf{z}^{[m]} - \mathbf{z}^{[m]}_{\hat{y}})_y - \text{ens}(\mathbf{z}^{[m]} - \mathbf{z}^{[m]}_{\hat{y}})_{\hat{y}}$$
$$= \text{ens}(k^{[m]}, \cdots)_y - \text{ens}(k^{[m]}, \cdots)_{\hat{y}} \triangleq h_m(k^{[m]}). \tag{8}$$

---

**Algorithm 1** The MORA white-box robust evaluation for ensemble defenses.

---

1: **function** MORA_Attack($f_{[1:M]}, \mathbf{x}, y, \beta, \tau, \nu, \epsilon, I$)
2:   $\hat{\mathbf{x}}_0 \leftarrow \mathcal{P}_{\epsilon,\mathbf{x}}\left(\mathbf{x} + \mathbf{u}\right)$, where $\mathbf{u} \sim \mathcal{U}\left(-\epsilon, \epsilon\right)$      ▷ Random init
3:   $\boldsymbol{\mu}_0 \leftarrow 0$
4:   **for** $i \in [0:I-1]$ **do**
5:     $\mathbf{z}^{[m]} \leftarrow f_m(\hat{\mathbf{x}}_i)$ for all $m \in [1:M]$      ▷ Sub-model predictions
6:     $\mathbf{z}^{[\mathrm{E}]} \leftarrow \frac{1}{M}\sum_{m\in[1:M]} \mathrm{ens}(\mathbf{z}^{[m]})$      ▷ Ensemble prediction
7:     $k^{[\mathrm{E}]} \leftarrow \mathbf{z}_y^{[\mathrm{E}]} - \mathbf{z}_{\hat{y}}^{[\mathrm{E}]}$
8:     **if** $k^{[\mathrm{E}]} \leq 0$ **then return** $\hat{\mathbf{x}}_i$      ▷ Successful attack
9:     $\boldsymbol{g}_{i+1} \leftarrow \mathrm{sign}(\nabla_{\hat{\mathbf{x}}_i}\, \mathcal{L}_{\beta,\tau}^{\mathrm{mora}}(\mathbf{z}^{[1:M]}, \mathbf{z}^{[\mathrm{E}]}, y))$      ▷ Sign-gradient with the MORA loss
10:     $\alpha \leftarrow \epsilon(1 + \cos(^{i\pi}/_I))$      ▷ Cosine step-size schedule
11:     $\boldsymbol{\mu}_{i+1} \leftarrow \mathcal{P}_{\epsilon,\mathbf{x}}\left(\boldsymbol{\mu}_i + \alpha\boldsymbol{g}_{i+1}\right)$      ▷ Iterative update
12:     $\hat{\mathbf{x}}_{i+1} \leftarrow \mathcal{P}_{\epsilon,\mathbf{x}}\left(\hat{\mathbf{x}}_i + \nu\left(\boldsymbol{\mu}_{i+1} - \hat{\mathbf{x}}_i\right) + (1-\nu)\left(\hat{\mathbf{x}}_i - \hat{\mathbf{x}}_{i-1}\right)\right)$      ▷ ... with momentum
13:   **end for**
14:   **return** $\hat{\mathbf{x}}_I$      ▷ Give up after $I$ iterations
15: **end function**

---

The weights are thus defined as follows:

$$\lambda_\tau^{[m]}\big(\mathbf{z}^{[m]}\big) = \frac{\partial k^{[\mathrm{E}]}(k^{[m]})}{\partial k^{[m]}} = \frac{\partial}{\partial k^{[m]}}\left(\frac{1}{M}\sum_{m\in[1:M]}\frac{\partial h_m(k^{[m]})}{\partial k^{[m]}}\right) = \frac{1}{M}\frac{\partial h_m(k^{[m]})}{\partial k^{[m]}}. \tag{9}$$

While it is possible to compute the weights using gradient back-propagation, we can simply derive the following closed-form solution of the weights for each of the three ensemble-forming strategies. For $\mathrm{wta}$, we use the softened version of $\mathrm{wta}$ as defined in (5) and can derive the weights as follows:

$$\lambda_\tau^{[m]}\big(\mathbf{z}^{[m]}\big) = 1[k^{[m]} > 0]\cdot\mathrm{detach}\big(\tfrac{1}{\tau M}\mathbf{s}_{\hat{y}}(1 + \mathbf{s}_y - \mathbf{s}_{\hat{y}})\big), \quad \text{where } \mathbf{s} = \mathrm{softmax}\big(\mathbf{z}^{[m]}/\tau\big). \tag{10}$$

Here $1[k^{[m]} > 0]$ is the indicator function that equals 1 if $k^{[m]} > 0$, or 0 otherwise, effectively stopping the attack on the $m^{\mathrm{th}}$ sub-model upon success, and the $\mathrm{detach}$ operator admits no backward propagation to its input. In the case of using sums of sub-model softmax outputs to form an ensemble decision, *i.e.*, $\mathrm{ens} = \mathrm{softmax}$, it is a special case of $\mathrm{softwta}_\tau$ where the temperature coefficient can be fixed at $\tau = 1$. Finally, when $\mathrm{ens} = \mathrm{id}$, *i.e.*, forming ensembles by summing logits, $\lambda_\tau^{[m]}(\mathbf{z}^{[m]})$ simply reduces to $1[k^{[m]} > 0]$ for the $m^{\mathrm{th}}$ sub-model.

### 3.3 The MORA Loss

For reference, defenses mechanisms we examine in this paper aim to find $\hat{\mathbf{x}}$ which maximizes the SCE loss $\mathcal{L}^{\mathrm{sce}}(\mathbf{z}^{[\mathrm{E}]}, y)$, to evaluate the ensemble robustness. The MORA loss improves this further by proposing two additional modifications to the untargeted loss function used to attack ensembles:

$$\mathcal{L}_{\beta,\tau}^{\mathrm{mora}}(\mathbf{z}^{[1:M]}, \mathbf{z}^{[\mathrm{E}]}, y) \triangleq \mathcal{L}^{\mathrm{sce}}\big(\beta\,\mathrm{norm}\big(\textstyle\sum_{m\in[1:M]}\lambda_\tau^{[m]}(\mathbf{z}^{[m]})\cdot\mathbf{z}^{[m]}\big) + (1-\beta)\,\mathrm{norm}(\mathbf{z}^{[\mathrm{E}]}), y\big). \tag{11}$$

First, it additionally introduces a sum of the $\lambda_\tau^{[m]}$-weighted variant of sub-model logits, in order to expose sub-model logits with adaptive reweighing described in Section 3.2. Second, $\beta$ interpolates the importance of the newly added auxiliary logits and the original ensemble logits. Finally, inspired by the effective surrogate loss in [32], it further normalizes the logits by their respective DL using:

$$\mathrm{norm}(\mathbf{z}) \triangleq 1[\mathbf{z}_y - \mathbf{z}_{\hat{y}} > 0]\cdot\mathbf{z}/\mathrm{detach}(\mathbf{z}_y - \mathbf{z}_{\hat{y}}). \tag{12}$$

Finally, the targeted variant of the MORA loss simply replaces $y$ with $t$ where $t$ is the intended target.

### 3.4 Improving the State-of-the-art

While the new $\mathcal{L}_{\beta,\tau}^{\mathrm{mora}}$ loss is highly effective against ensemble defenses we test in this paper, we strive for further advances in MORA's ability to generate faster and better adversarial examples. Inspired by recent publications, we borrow ideas from related adversarial attack tactics, which includes adopting a cosine step-size schedule [16], momentum [7, 5], random restarts [25] and multiple target attacks [5, 25]. We provide the overall algorithm in Algorithm 1, which computes an adversarial image $\hat{\mathbf{x}}_I$ as return, by taking as input the sub-models $f_{[1:M]}$, natural image $\mathbf{x}$, ground truth label $y$, $\beta$ to interpolate between the auxiliary logits and the original, $\tau$ controls the temperature, momentum $\mu = 0.75$ following [32, 5], $\epsilon$ perturbation bound, and finally the maximum number of iterations $I$.

Table 1: Comparing accuracies among iterative methods [17, 3], learned attacks (**CAA** [18]), AutoAttack (**AA**) [5], adaptive auto attack (**A**$^3$) [16], and MORA across various ensemble defense strategies under 3 ensembling modes (softmax, voting and logits), and $\epsilon = 0.01$. The "Complexity" row shows the worst-case complexity in iteration counts. The "$\Delta$" column shows the accuracy overestimation from self-reported/reproduced "**Nominal**" values to MORA$^{mt}$. Baselines with † are reproduced with source code. All results are re-run 5 times and within $\pm0.05\%$ standard deviation.

| Defense Complexity | # | Clean 1 | Nominal — | PGD 500 | CW 500 | MORA 500 | A$^3$ 12k | AA 4.9k | CAA 1.8k | MORA$^{mt}$ 1.4k | Δ |
|---|---|---|---|---|---|---|---|---|---|---|---|
| **Softmax** | | | | | | | | | | | |
| ADP | 3 | 92.88 | 29.12 | 5.98 | 7.72 | 0.59 | 2.12 | 0.98 | 3.34 | **0.34** | 28.78 |
| | 5 | 93.34 | 25.14 | 7.10 | 8.70 | 0.97 | 3.62 | 2.18 | 4.25 | **0.67** | 24.47 |
| | 8 | 93.48 | 20.20 | 9.22 | 9.59 | 1.70 | 4.84 | 3.94 | 6.04 | **1.32** | 18.88 |
| Dverge | 3 | 91.99 | 47.42 | 44.49 | 40.17 | 25.77 | 33.36 | 30.58 | 32.98 | **25.26** | 22.16 |
| | 5 | 92.38 | 55.72 | 54.61 | 52.83 | 40.02 | 48.41 | 43.29 | 46.65 | **39.50** | 16.22 |
| | 8 | 91.65 | 59.63 | 59.13 | 58.25 | 55.68 | 57.29 | 56.71 | 56.89 | **55.57** | 4.06 |
| GAL | 3 | 89.41 | 19.48 | 8.13 | 11.57 | 0.67 | 0.70 | 0.85 | 1.00 | **0.51** | 18.97 |
| | 5 | 90.93 | 41.38 | 37.59 | 35.52 | 17.45 | 26.94 | 23.90 | 25.11 | **16.05** | 25.33 |
| | 8 | 92.45 | 56.31 | 53.39 | 52.56 | 28.71 | 36.51 | 37.46 | 35.30 | **27.44** | 28.87 |
| TRS† | 3 | 70.02 | 19.71 | 14.01 | 10.87 | 8.11 | 8.72 | 8.46 | 9.75 | **7.60** | 12.11 |
| | 5 | 69.00 | 23.17 | 15.91 | 15.28 | 12.67 | 13.22 | 13.20 | 13.78 | **12.47** | 10.70 |
| | 8 | 73.01 | 23.64 | 18.02 | 17.59 | 15.90 | 16.22 | 16.51 | 16.73 | **15.64** | 8.00 |
| **Voting** | | | | | | | | | | | |
| ADP | 3 | 91.84 | 41.62† | 9.32 | 11.84 | 0.64 | 3.06 | 6.13 | 8.29 | **0.29** | 41.33 |
| | 5 | 93.13 | 40.29† | 12.42 | 12.05 | 1.17 | 6.03 | 10.13 | 0.67 | **0.62** | 39.67 |
| | 8 | 93.28 | 30.10† | 12.53 | 10.50 | 3.16 | 6.11 | 9.21 | 1.69 | **1.65** | 28.45 |
| Dverge | 3 | 91.72 | 39.05† | 31.48 | 28.00 | 23.57 | 24.95 | 24.98 | 27.65 | **22.91** | 16.14 |
| | 5 | 92.18 | 49.36† | 44.28 | 42.28 | 35.06 | 39.15 | 39.20 | 40.85 | **34.46** | 14.90 |
| | 8 | 91.58 | 56.85† | 53.72 | 52.35 | 47.12 | 50.58 | 50.04 | 51.15 | **46.10** | 10.75 |
| GAL | 3 | 89.09 | 21.48† | 5.85 | 7.64 | 0.87 | 0.71 | 0.56 | 0.78 | **0.35** | 21.13 |
| | 5 | 90.77 | 37.32† | 29.33 | 27.62 | 12.96 | 18.55 | 20.82 | 22.17 | **12.25** | 25.07 |
| | 8 | 92.37 | 55.39† | 49.56 | 48.02 | 21.66 | 30.35 | 31.39 | 30.93 | **20.16** | 35.23 |
| TRS† | 3 | 68.95 | 13.79 | 10.19 | 8.71 | 5.73 | 11.89 | 6.69 | 8.08 | **5.44** | 8.35 |
| | 5 | 68.31 | 15.36 | 12.71 | 11.88 | 8.82 | 10.08 | 10.30 | 11.21 | **8.38** | 6.98 |
| | 8 | 72.05 | 17.00 | 14.57 | 13.48 | 11.39 | 11.99 | 11.85 | 12.80 | **10.69** | 6.31 |
| **Logits** | | | | | | | | | | | |
| ADP | 3 | 92.86 | 3.44† | 0.87 | 2.05 | 0.48 | 0.25 | 0.22 | 0.31 | **0.21** | 3.23 |
| | 5 | 93.48 | 4.57† | 1.97 | 4.24 | 1.12 | 1.00 | 0.97 | 1.09 | **0.89** | 3.68 |
| | 8 | 93.38 | 5.39† | 3.57 | 4.77 | 2.13 | 2.20 | 2.05 | 2.11 | **1.93** | 3.46 |
| Dverge | 3 | 92.19 | 38.31† | 37.99 | 38.60 | 36.89 | 36.94 | 36.96 | 37.07 | **36.84** | 1.47 |
| | 5 | 92.28 | 50.77† | 50.57 | 51.28 | 49.65 | 49.72 | 49.66 | 49.75 | **49.59** | 1.18 |
| | 8 | 91.73 | 61.06† | 60.95 | 61.51 | 60.52 | 60.59 | 60.52 | 60.55 | **60.49** | 0.57 |
| GAL | 3 | 89.50 | 15.47† | 10.01 | 10.53 | 0.52 | **0.02** | **0.02** | 0.08 | 0.03 | 15.44 |
| | 5 | 90.93 | 36.36† | 33.97 | 35.14 | 22.24 | 33.43 | 20.24 | 21.66 | **19.40** | 16.96 |
| | 8 | 92.54 | 56.08† | 53.67 | 54.69 | 31.52 | 40.90 | 30.89 | 31.17 | **30.66** | 25.42 |
| TRS† | 3 | 69.72 | 13.31 | 13.06 | 13.80 | 12.11 | 12.13 | 12.16 | 12.21 | **12.07** | 1.24 |
| | 5 | 68.90 | 16.89 | 16.65 | 17.34 | 15.88 | 15.86 | 15.90 | 15.95 | **15.82** | 1.07 |
| | 8 | 72.24 | 19.40 | 19.20 | 19.67 | 18.20 | 18.18 | 18.27 | 18.34 | **18.17** | 1.23 |

## 4 Experimental Results

We compare MORA against SOTA attacks for a wide range of ensemble defenses under three ensemble-forming strategies (softmax, voting, and logits). We use pre-trained ResNet-20 [12] models from ADP [19], Dverge [30], GAL [14], and reproduced TRS [31] using the same architecture with official source code, as pre-trained models were unavailable. Our robustness evaluation considers the $\ell^\infty$ white-box attacks on the CIFAR-10 test set [15], with perturbation $\epsilon = 0.01$ unless specified. The full comparison results can be found in Table 1; larger $\epsilon$ comparisons, and similar results on CIFAR-100 models are in Appendix A. We provide our key observations below.

**Traditional attacks may fail to break through gradient obfuscation.** We reproduce two traditional white-box attacks, *i.e.*, projected gradient descent (PGD) [17] and C&W [3] with 5 random restarts, each with a maximum of 100 iterations, giving a total of 500 iterations. PGD uses a fixed step size of $\epsilon/4$. For a fair comparison, MORA with 500 iterations sweeps $\beta \in \{0, 0.25, 0.5, 0.75, 1\}$, with each $\beta$ up to 100 iterations. Even with a 500 iteration budget, it is clear that PGD and C&W may substantially overestimate robustness, especially when tested under the softmax and voting ensemble-forming options, and MORA can work around this obstacle thanks to its attacks on sub-models.

**Diversified gradients can hamper even integrated attacks with large arsenals.** Moreover, we test the defenses against recent integrated attacks with SOTA baselines on robustness evaluation, namely Adaptive Auto Attack ($\text{A}^3$) [16], AutoAttack (AA) [5], and Composite Adversarial Attacks (CAA) [18], which comprise large arsenals of various attack strategies. We reproduce CAA following [18] to search for the attack policy before evaluating the defending models. Note that its computational complexity is thus much higher than the other attacks, but we only report its test-time complexity. In particular, while they enjoyed much higher success rates than PGD and C&W, some defenses render their tactics ineffective. We observe, *e.g.*, sizeable robustness overestimation on ADP [19] under softmax and voting, which explicitly diversifies sub-model gradients. As MORA can dynamically re-adjust sub-model importance *w.r.t.* their "ease-of-attack", it performs substantially better with much fewer iterations. In addition to the earlier 500 iterations, the multi-targeted MORA$^{\text{mt}}$ targets the remaining 9 class labels with 100 iterations for each label and $\beta$ fixed at $0.5$. Others also use multi-targeted attacks along with respective tactics.

**Robustness of most sub-models *vs.* robustness of ensemble.** We find that robustness of a majority of sub-models (fooling $3/8$ for softmax and $2/8$ for logits) usually do not translate to the overall robustness of the ensemble (Figures 1c and 3a). As voting requires breaking $1/2$ sub-models simultaneously (Figure 3b), it is perceivable that using voting may give rise to a higher overall robustness. Yet surprisingly, for most defending ensembles, voting performs worse than softmax and logits.

**Ensemble-forming strategies may give a false sense of security.** On one hand, softmax and voting strategies exhibit substantially larger overestimated robustness (up to 40%) than logits. On the other hand, in stark contrast to the proposed use of softmax from [14, 19, 30, 31], we find summing by logits can form ensembles that are notably more robust than the other two (Figure 3c), while attackers only needs to successfully deceive a few sub-models (referring back to Figure 1c).

**Up to $60\times$ faster convergence** under 500 iterations. Figure 4 compares the convergence speed of MORA against AA losses, C&W, and PGD on defending ensembles. MORA converges substantially faster than the other attacks, using only up to 31 steps to match AA losses with $500$ iterations.

**Ensemble defense mechanisms may be at odds with robustness.** In Table 2, we compare respective attacks on adversarially trained Dverge models. To our surprise, forming larger ensembles is actually harmful to the robustness of ensemble.

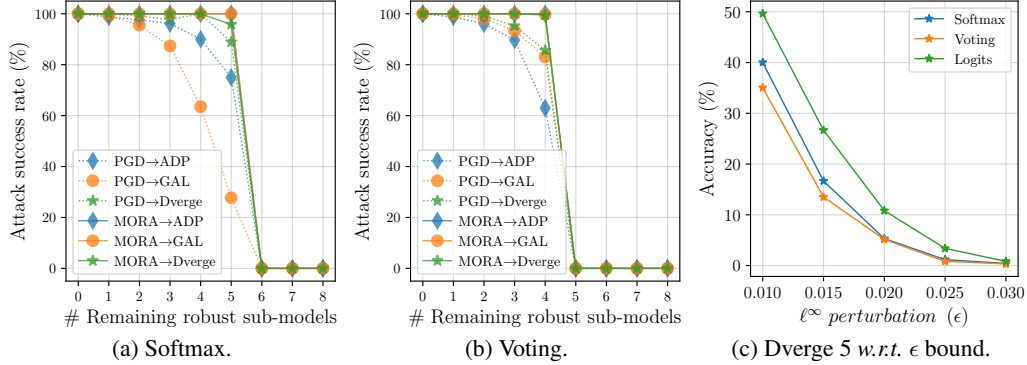

(a) Softmax.       (b) Voting.       (c) Dverge 5 *w.r.t.* $\epsilon$ bound.

Figure 3: (a, b) MORA can successfully fool ensemble-forming methods (ADP [19], GAL [14] and Dverge [30]) even with the majority of their sub-models giving correct outputs under softmax and logits (Figure 1c). While voting requires $\geq 1/2$ sub-models to be incorrect, it is unfortunately the least robust option in all defenses. "$A \rightarrow B$" means using $A$ to attack $B$ for up to 100 iterations. (c) Dverge with 5 sub-models *w.r.t.* the $\epsilon$ bound on $\ell^\infty$ perturbation. Contrary to existing literatures, we find logits to be the most robust option of the three ensemble-forming strategies.

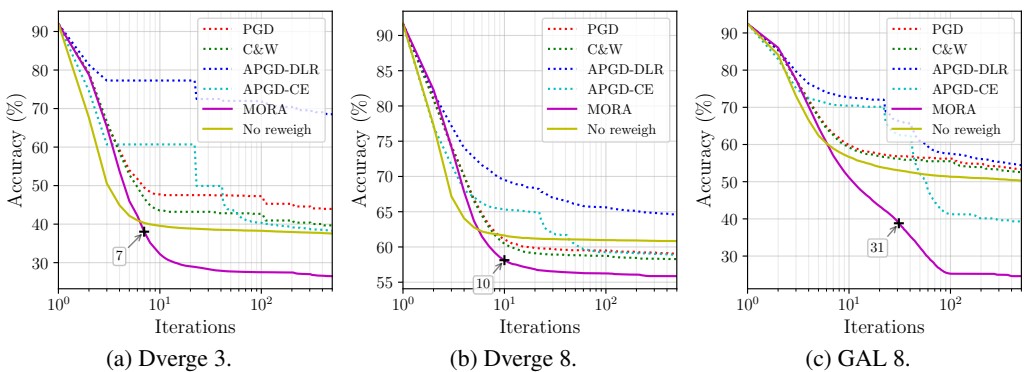

(a) Dverge 3.  (b) Dverge 8.  (c) GAL 8.

Figure 4: (a, b, c) Comparing the convergence speed of MORA against a variant of MORA without sub-model reweighing, C&W, PGD and AA losses (APGD-{DLR,CE}) on Dverge and GAL with softmax. While "No reweigh" converge faster initially, it struggles to improve after 10 iterations; in contrast, adaptive reweighing (MORA) continues to converge at a much faster rate. The horizontal and vertical axes respectively show the iteration count used, and the percentage of unsuccessful images remaining. We annotated the number of iterations for MORA to overtake all with 500 iterations.

Table 2: Attacking adversarially trained Dverge [30] models under the same setting as Table 1, except we let $\epsilon = 0.03$. Notably, forming larger ensembles can actually be detrimental to robustness.

| Dverge Complexity | # | Clean 1 | Nominal — | PGD 500 | CW 500 | MORA 500 | $A^3$ 12k | AA 4.9k | CAA 1.8k | MORA$^{mt}$ 1.4k | $\Delta$ |
|---|---|---|---|---|---|---|---|---|---|---|---|
| Softmax | 3 | 83.78 | 45.09 | 44.85 | 44.21 | 42.91 | 42.66 | 42.70 | 42.69 | **42.65** | 2.44 |
| | 5 | 86.09 | 42.57 | 42.40 | 42.51 | 41.05 | **40.74** | 40.85 | 40.84 | 40.75 | 1.79 |
| | 8 | 86.69 | 40.80 | 40.58 | 40.94 | 39.49 | **39.33** | 39.40 | 39.39 | 39.35 | 1.41 |
| Voting | 3 | 83.67 | 59.13† | 55.35 | 55.85 | 38.59 | 38.78 | 39.97 | 40.31 | **38.24** | 20.89 |
| | 5 | 86.05 | 47.72† | 44.32 | 45.18 | 36.19 | 36.42 | 37.29 | 37.89 | **36.01** | 11.71 |
| | 8 | 86.54 | 38.93† | 37.16 | 38.17 | 34.20 | 34.89 | 35.54 | 36.19 | **34.03** | 4.90 |
| Logits | 3 | 83.74 | 44.83† | 44.69 | 44.24 | 42.83 | 42.66 | 42.70 | 42.70 | **42.63** | 2.20 |
| | 5 | 86.03 | 42.47† | 42.22 | 42.63 | 41.00 | 40.91 | 40.92 | 40.92 | **40.87** | 1.60 |
| | 8 | 86.65 | 40.53† | 40.33 | 41.14 | 39.53 | **39.43** | 39.50 | 39.47 | **39.43** | 1.09 |

**Additional results, ablation, and sensitivity analyses.** Due to the page limit, we provide full results of relevant figures in Appendix A, note that the above key observations still hold true for all ensemble defenses we test under different ensemble-forming strategies and $\epsilon$ perturbation bounds. In addition, we provide extensive ablation study on the design choices we made, and sensitivity analysis on the temperature constant $\tau$.

## 5 Conclusions

This paper identifies severe robustness overestimation in many ensemble defense techniques, and further investigates problem the robustness evaluation under three ensemble-forming strategies. To efficiently and accurately evaluate the robustness of ensembles, we introduce MORA, a new attack technique which reweighs sub-model importance adaptively by their respective "ease-of-attack" during attack iterations. MORA enjoys a much improved success rate and convergence rate compared with other SOTA attacks. Moreover, we found several surprising observations related to ensemble defenses, for instance, (1) misleading a minority of sub-models is sufficient to fool the ensemble, (2) summing by logits is the simplest yet most robust way to form ensembles, (3) with adversarial training, ensemble defenses may actually harm robustness, *etc*. We hope the above observations may help to guide future avenue on ensemble defenses, and provide a strong attack baseline for potential approaches. Finally, MORA is open source with reproducible results and pre-trained models; and we continually update a leaderboard of ensemble defenses under various attack strategies.

## Acknowledgements

This work is supported in part by National Key R&D Program of China (No. 2019YFB2102100), Key-Area Research and Development Program of Guangdong Province (No. 2020B010164003), Science and Technology Development Fund of Macao S.A.R (FDCT) under No. 0015/2019/AKP, Shenzhen Science and Technology Innovation Commission (No. JCYJ20190812160003719), and Shenzhen Industrial Application Projects of undertaking the National Key R&D Program of China (No.CJGJZD20210408091600002). This work was carried out in part at SICC which is supported by SKL-IOTSC, University of Macau.

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
