}}(\mathbf{x} + \mathbf{u})$, where $\mathbf{u} \sim \mathcal{U}(-\epsilon, \epsilon)$ $\hfill \triangleright$ Random init
3: $\quad \boldsymbol{\mu}_0 \leftarrow 0$
4: $\quad$ **for** $i \in [0 : I-1]$ **do**
5: $\qquad \mathbf{z}^{[m]} \leftarrow f_m(\hat{\mathbf{x}}_i)$ for all $m \in [1:M]$ $\hfill \triangleright$ Sub-model predictions
6: $\qquad \mathbf{z}^{[\mathrm{E}]} \leftarrow \frac{1}{M} \sum_{m \in [1:M]} \mathrm{ens}(\mathbf{z}^{[m]})$ $\hfill \triangleright$ Ensemble prediction
7: $\qquad k^{[\mathrm{E}]} \leftarrow \mathbf{z}_y^{[\mathrm{E}]} - \mathbf{z}_{\hat{y}}^{[\mathrm{E}]}$
8: $\qquad$ **if** $k^{[\mathrm{E}]} \leq 0$ **then return** $\hat{\mathbf{x}}_i$ $\hfill \triangleright$ Successful attack
9: $\qquad \boldsymbol{g}_{i+1} \leftarrow \mathrm{sign}(\nabla_{\hat{\mathbf{x}}_i} \mathcal{L}_{\beta,\tau}^{\mathrm{mora}}(\mathbf{z}^{[1:M]}, \mathbf{z}^{[\mathrm{E}]}, y))$ $\hfill \triangleright$ Sign-gradient with the MORA loss
10: $\qquad \alpha \leftarrow \epsilon(1 + \cos(i\pi/I))$ $\hfill \triangleright$ Cosine step-size schedule
11: $\qquad \boldsymbol{\mu}_{i+1} \leftarrow \mathcal{P}_{\epsilon,\mathbf{x}}(\boldsymbol{\mu}_i + \alpha \boldsymbol{g}_{i+1})$ $\hfill \triangleright$ Iterative update
12: $\qquad \hat{\mathbf{x}}_{i+1} \leftarrow \mathcal{P}_{\epsilon,\mathbf{x}}(\hat{\mathbf{x}}_i + \nu(\boldsymbol{\mu}_{i+1} - \hat{\mathbf{x}}_i) + (1-\nu)(\hat{\mathbf{x}}_i - \hat{\mathbf{x}}_{i-1}))$ $\hfill \triangleright \ldots$ with momentum
13: $\quad$ **end for**

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

# A Additional Results

## A.1 Larger Perturbations

Tables 3 and 4 compare the effectiveness of iterative methods [17, 3], learned attacks (**CAA** [18]), AutoAttack (**AA**) [5], MORA and MORA[mt] across various ensemble defense strategies with ResNet-20 [12] sub-models under 3 ensemble-forming modes (softmax, voting and logits). Here, Table 3 uses an $\ell^\infty$ perturbation bound $\epsilon = 0.02$, whereas Table 4 uses $\epsilon = 0.03$. All results are re-run 5 times and within $\pm 0.05\%$ standard deviation. Note that under $\epsilon = 0.02$ most defense methods only retain up to 4% robust accuracies. Dverge [30] with 8 sub-models however surprisingly has $23.31\%$ robustness against the strongest MORA[mt] attack. We believe this can be attributed to the fact that similarly to adversarial training, Dverge diversifies sub-models with adversarial examples from each other, rather than via explicit regularization as carried out by the other defenses. It thus requires substantially higher training cost than the others ($\sim 5\times$ ADP [19]).

Finally, Figure 5 compares the different ensemble-forming strategies under an increasing $\ell^\infty$ perturbation bound $\epsilon \in \{0.01, 0.015, 0.02, 0.025, 0.03\}$. Note that across all configurations, logit-based ensembles are the most robust.

## A.2 Convergence Speed Comparisons

Figures 6 and 7 compare the convergence speed of MORA against SOTA attacks for up to 500 iterations. As AA comprises multiple attack strategies, we extract its two gradient-based attacks (APGD-{CE,DLR}) to facilitate comparisons. Each attack uses 5 restarts, each with up to 100 iterations. The horizontal axes show the iteration counts used, and the vertical axes denote the percentages of remaining unsuccessful images. When forming ensembles with logits, MORA and "No reweighing" are both identical. Note that MORA substantially outperforms most existing attacks under 500 iterations, and generally requires (up to $70\times$) fewer iterations to achieve the same accuracies of other attacks with 500 iterations.

## A.3 Ablation and Sensitivity Analyses

In Table 5, we perform ablation of the individual components used in MORA. We begin with the standard "PGD" attack [17] with 5 random restarts, each with 100 iterations, and a constant step size of $\epsilon/4$. Each row then consecutively adds a new component. "Momentum" introduces momentum $\nu = 0.75$ as used in Algorithm 1, "Cosine Step Size" then replaces the constant step size with a cosine schedule $\alpha_i = \epsilon(1 + \cos(i\pi/I))$. "Sub-model Logits" further exploits the sub-model logits directly following Section 3.1, and replaces random restarts with a $\beta \in \{0, 0.25, 0.5, 0.75, 1\}$ grid search to match the cost of 500 iterations. "Logit Normalization" incorporates the normalization of logits as proposed in (12) of Section 3.3. "Adaptive Reweighing" adaptively adjusts sub-model weights for attacking ensemble defenses ((10) of Section 3.2). Finally, "Multiple Targets" [11] additionally uses 100 iterations of the targeted variant of $\mathcal{L}_{\beta,\tau}^{\mathrm{mora}}$ on each of the remaining 9 class labels (Section 3.3).

Increasing the temperature coefficient $\tau$ from 1 as used in (11) can affect the convergence speed of MORA. To ensure a fair comparison in our results, we fix a constant $\tau = 5$ for softmax-based ensembles, and $\tau = 10$ for voting, as increasing $\tau$ may help improve convergence. In addition, and Figure 8 provides sensitivity analyses of $\tau$ on the three defending methods (ADP, GAL, Dverge).

Finally, Figure 9 shows the effect of varying $\beta \in [0, 1]$, the interpolation between $\mathcal{L}_{\beta,\tau}^{\mathrm{mora}}$ and the ensemble's original loss $\mathcal{L}^{\mathrm{sce}}$. Introducing $\mathcal{L}_{\beta,\tau}^{\mathrm{mora}}$ substantially improves the strength of attack. Note that in our comparison results (Tables 1, 2, 3, 4 and 6), instead of 5 random restarts we use a $\beta \in \{0, 0.25, 0.5, 0.75, 1\}$ schedule to further improve the final attack success rate.

## A.4 CIFAR-100

Table 6 compares the attacks on PDD+DEG [13] defenses trained on CIFAR-100. Similar to GAL [14] and TRS [31], PDD+DEG also diversifies sub-models gradients by minimizing cosine-similarities of gradients via regularization, and further diversifies feature selection with adaptive dropouts. We report attacks on ensembles with three ensemble-forming methods (softmax, voting and logits). Note that we only included PDD+DEG as other methods did not train on CIFAR-100.

Table 3: Comparing the accuracies of SOTA attacks and MORA on various defenses. Please refer to Table 1 for a detailed explanation. This table uses $\epsilon = 0.02$ as the $\ell^\infty$ perturbation bound, under which most defenses (27/48) have close to nil robustness ($\leq 2\%$).

| Defense Complexity | # | Clean 1 | Nominal — | PGD 500 | CW 500 | MORA 500 | AA 4.9k | CAA 1.8k | MORA^mt 1.4k | Δ |
|---|---|---|---|---|---|---|---|---|---|---|
| **Softmax** | | | | | | | | | | |
| ADP | 3 | 92.88 | 12.53 | 0.07 | 0.23 | **0.00** | **0.00** | 0.03 | **0.00** | 12.53 |
| | 5 | 93.34 | 12.75 | 0.31 | 0.49 | **0.00** | 0.01 | 0.10 | **0.00** | 12.75 |
| | 8 | 93.48 | 12.61 | 3.04 | 1.68 | **0.00** | 0.02 | 0.47 | **0.00** | 12.61 |
| Dverge | 3 | 91.99 | 12.78 | 10.16 | 8.09 | 0.95 | 6.33 | 6.77 | **0.78** | 12.00 |
| | 5 | 92.38 | 22.36 | 20.33 | 18.88 | 5.31 | 10.12 | 16.13 | **4.73** | 17.63 |
| | 8 | 91.65 | 28.20 | 26.66 | 26.65 | 17.29 | 22.98 | 25.40 | **16.94** | 11.26 |
| GAL | 3 | 89.41 | 1.75 | 0.12 | 0.35 | **0.00** | **0.00** | **0.00** | **0.00** | 1.75 |
| | 5 | 90.93 | 6.80 | 4.81 | 4.22 | 0.64 | 2.87 | 2.77 | **0.49** | 6.31 |
| | 8 | 92.45 | 12.22 | 10.75 | 10.80 | 3.43 | 8.71 | 6.94 | **2.97** | 9.25 |
| TRS† | 3 | 70.02 | 2.24 | 1.49 | 0.80 | 0.20 | 0.33 | 0.86 | **0.18** | 2.06 |
| | 5 | 69.00 | 3.56 | 1.91 | 1.75 | 0.72 | 1.07 | 1.68 | **0.64** | 2.92 |
| | 8 | 73.01 | 4.18 | 1.71 | 1.77 | 1.14 | 1.50 | 1.72 | **0.97** | 3.21 |
| **Voting** | | | | | | | | | | |
| ADP | 3 | 91.84 | 22.18† | 0.35 | 0.39 | **0.00** | 0.19 | 0.41 | **0.00** | 22.18 |
| | 5 | 93.13 | 21.76† | 0.80 | 0.96 | **0.00** | 0.58 | 0.67 | **0.00** | 21.76 |
| | 8 | 93.28 | 15.19† | 3.47 | 2.07 | 0.02 | 0.90 | 1.15 | **0.00** | 15.19 |
| Dverge | 3 | 91.72 | 4.70† | 2.61 | 1.54 | 1.11 | 1.47 | 2.38 | **0.89** | 3.81 |
| | 5 | 92.18 | 9.30† | 7.68 | 6.38 | 5.15 | 7.15 | 8.33 | **4.84** | 4.46 |
| | 8 | 91.58 | 18.15† | 16.82 | 15.81 | 11.99 | 15.56 | 17.16 | **11.47** | 6.68 |
| GAL | 3 | 89.09 | 3.57† | 0.23 | 0.11 | **0.00** | **0.00** | **0.00** | **0.00** | 3.57 |
| | 5 | 90.77 | 3.26† | 1.95 | 1.83 | 0.54 | 1.64 | 2.10 | **0.48** | 2.78 |
| | 8 | 92.37 | 9.36† | 7.26 | 6.85 | 2.34 | 4.91 | 5.33 | **1.90** | 7.46 |
| TRS† | 3 | 68.95 | 0.83 | 0.53 | 0.25 | 0.16 | 0.59 | 0.34 | **0.12** | 0.71 |
| | 5 | 68.31 | 1.34 | 0.92 | 0.83 | 0.60 | 0.72 | 0.91 | **0.53** | 0.81 |
| | 8 | 72.05 | 1.45 | 1.17 | 0.87 | 0.86 | 0.71 | 1.00 | **0.68** | 0.77 |
| **Logits** | | | | | | | | | | |
| ADP | 3 | 92.86 | 0.13† | **0.00** | **0.00** | **0.00** | **0.00** | **0.00** | **0.00** | 0.13 |
| | 5 | 93.48 | 0.20† | **0.00** | 0.04 | **0.00** | **0.00** | **0.00** | **0.00** | 0.20 |
| | 8 | 93.38 | 0.04† | **0.00** | **0.00** | **0.00** | **0.00** | **0.00** | **0.00** | 0.04 |
| Dverge | 3 | 92.19 | 4.39† | 4.12 | 4.24 | 3.27 | 3.31 | 3.40 | **3.17** | 1.22 |
| | 5 | 92.28 | 12.66† | 12.24 | 12.56 | 10.83 | 10.93 | 10.99 | **10.80** | 1.86 |
| | 8 | 91.73 | 24.75† | 24.41 | 25.42 | 23.32 | 23.39 | 23.49 | **23.31** | 1.54 |
| GAL | 3 | 89.50 | 1.10† | 0.25 | 0.22 | **0.00** | **0.00** | **0.00** | **0.00** | 1.10 |
| | 5 | 90.93 | 3.98† | 3.14 | 3.93 | 1.01 | 0.47 | 0.64 | **0.45** | 3.53 |
| | 8 | 92.54 | 11.18† | 9.81 | 10.81 | 3.86 | 3.52 | 3.79 | **3.41** | 7.77 |
| TRS† | 3 | 69.72 | 0.73 | 0.64 | 0.71 | 0.48 | 0.50 | 0.52 | **0.47** | 0.26 |
| | 5 | 68.90 | 1.60 | 1.51 | 1.63 | 1.35 | 1.31 | 1.33 | **1.30** | 0.30 |
| | 8 | 72.24 | 1.78 | 1.69 | 1.78 | 1.52 | 1.47 | 1.46 | **1.42** | 0.36 |

## A.5  Failure Modes in Ensemble Defenses

Figure 10 provides a visualization of the loss surfaces of ADP [19] under 3 different ensemble-forming strategies. In this section, we continue the discussion of the two failure modes in ensemble defense that induce overestimated robustness as introduced in Section 1:

(a) *Gradient obfuscation via ensemble-forming strategies.* It is evident that under PGD-10 attacks, both softmax- and voting-based ensembles (Figures 10a and 10b respectively) exhibit to some extent gradient obfuscation as they result in flatter loss surfaces in the adversarial direction $g$, whereas the logits-based variant does not (Figure 10c).

(b) *Gradient diversification.* As sub-model gradients counteract, PGD attacks on softmax- and voting-based ensembles may result in an averaged gradient direction $g$ that experience difficulty in

Table 4: Comparing the accuracies of SOTA attacks and MORA on various defenses. Please refer to Table 1 for a detailed explanation. This table uses $\epsilon = 0.03$ as the $\ell^\infty$ perturbation bound, under which almost all defenses exhibit $0\%$ robustness, and all defenses fail to give $5\%$ robustness.

| Defense Complexity | # | Clean 1 | Nominal — | PGD 500 | CW 500 | MORA 500 | AA 4.9k | CAA 1.8k | MORA$^{mt}$ 1.4k | Δ |
|---|---|---|---|---|---|---|---|---|---|---|
| | | | | | **Softmax** | | | | | |
| ADP | 3 | 92.88 | 7.29 | **0.00** | **0.00** | **0.00** | **0.00** | **0.00** | **0.00** | 7.29 |
| | 5 | 93.34 | 8.52 | 0.04 | 0.03 | **0.00** | **0.00** | **0.00** | **0.00** | 8.52 |
| | 8 | 93.48 | 9.64 | 1.75 | 0.89 | **0.00** | **0.00** | 0.05 | **0.00** | 9.64 |
| Dverge | 3 | 91.99 | 2.91 | 1.43 | 1.47 | 0.02 | 1.59 | 1.10 | **0.00** | 2.91 |
| | 5 | 92.38 | 5.90 | 4.69 | 4.27 | 1.13 | 2.98 | 4.05 | **0.23** | 5.67 |
| | 8 | 91.65 | 9.17 | 8.28 | 8.46 | 2.54 | 6.84 | 8.21 | **2.34** | 6.83 |
| GAL | 3 | 89.41 | 1.26 | 0.01 | 0.06 | **0.00** | **0.00** | **0.00** | **0.00** | 1.26 |
| | 5 | 90.93 | 0.68 | 0.34 | 0.22 | **0.00** | 0.20 | 0.10 | **0.00** | 0.68 |
| | 8 | 92.45 | 1.18 | 0.92 | 0.88 | 0.27 | 1.19 | 0.79 | **0.17** | 1.01 |
| TRS† | 3 | 70.02 | 0.71 | 0.11 | 0.04 | **0.00** | 0.01 | 0.06 | **0.00** | 0.71 |
| | 5 | 69.00 | 2.24 | 0.18 | 0.17 | 0.04 | **0.03** | 0.14 | **0.03** | 2.21 |
| | 8 | 73.01 | 2.32 | 0.12 | 0.17 | 0.06 | 0.09 | 0.15 | **0.02** | 2.30 |
| | | | | | **Voting** | | | | | |
| ADP | 3 | 91.84 | 15.18† | 0.03 | 0.09 | **0.00** | 0.02 | 0.03 | **0.00** | 15.18 |
| | 5 | 93.13 | 14.97† | 0.33 | 0.17 | **0.00** | 0.03 | 0.04 | **0.00** | 14.97 |
| | 8 | 93.28 | 11.26† | 1.91 | 1.01 | **0.00** | 0.09 | 0.17 | **0.00** | 11.26 |
| Dverge | 3 | 91.72 | 0.99† | 0.19 | 0.07 | 0.06 | 0.05 | 0.10 | **0.00** | 0.99 |
| | 5 | 92.18 | 0.98† | 0.55 | 0.44 | 0.33 | 0.50 | 0.71 | **0.23** | 0.75 |
| | 8 | 91.58 | 3.42† | 2.72 | 2.27 | 1.72 | 2.30 | 2.92 | **1.56** | 1.86 |
| GAL | 3 | 89.09 | 1.90† | 0.02 | 0.04 | **0.00** | **0.00** | **0.00** | **0.00** | 1.90 |
| | 5 | 90.77 | 0.27† | 0.10 | 0.07 | **0.00** | 0.04 | 0.07 | **0.00** | 0.27 |
| | 8 | 92.37 | 0.66† | 0.44 | 0.38 | 0.15 | 0.36 | 0.45 | **0.13** | 0.53 |
| TRS† | 3 | 68.95 | 0.03 | 0.02 | **0.00** | **0.00** | **0.00** | **0.00** | **0.00** | 0.03 |
| | 5 | 68.31 | 0.16 | 0.10 | 0.04 | 0.06 | 0.01 | 0.03 | **0.00** | 0.09 |
| | 8 | 72.05 | 0.15 | **0.00** | 0.01 | **0.00** | **0.00** | **0.00** | **0.00** | 0.15 |
| | | | | | **Logits** | | | | | |
| ADP | 3 | 92.86 | 0.01† | **0.00** | **0.00** | **0.00** | **0.00** | **0.00** | **0.00** | 0.01 |
| | 5 | 93.48 | 0.05† | **0.00** | **0.00** | **0.00** | **0.00** | **0.00** | **0.00** | 0.05 |
| | 8 | 93.38 | **0.00**† | **0.00** | **0.00** | **0.00** | **0.00** | **0.00** | **0.00** | 0.00 |
| Dverge | 3 | 92.19 | 0.18† | 0.15 | 0.15 | **0.09** | **0.09** | 0.09 | **0.09** | 0.09 |
| | 5 | 92.28 | 1.25† | 1.13 | 1.13 | 0.83 | **0.79** | 0.83 | **0.79** | 0.46 |
| | 8 | 91.73 | 6.01† | 5.77 | 5.95 | 4.51 | 4.54 | 4.63 | **4.42** | 1.59 |
| GAL | 3 | 89.50 | 0.09† | **0.00** | 0.02 | **0.00** | **0.00** | **0.00** | **0.00** | 0.09 |
| | 5 | 90.93 | 0.24† | 0.16 | 0.13 | **0.00** | **0.00** | **0.00** | **0.00** | 0.24 |
| | 8 | 92.54 | 0.66† | 0.55 | 0.70 | 0.22 | **0.17** | 0.18 | **0.17** | 0.48 |
| TRS† | 3 | 69.72 | 0.01 | 0.01 | 0.02 | **0.00** | 0.01 | 0.01 | **0.00** | 0.01 |
| | 5 | 68.90 | 0.11 | 0.10 | 0.12 | 0.09 | 0.07 | 0.06 | **0.04** | 0.07 |
| | 8 | 72.24 | 0.13 | 0.12 | 0.15 | 0.07 | 0.08 | 0.07 | **0.06** | 0.07 |

increasing loss (Figures 10a and 10b respectively). Adopting sub-model reweighing (bottom row in Figure 10) alleviates this difficulty, and allows the attack to succeed more reliably.

# B   Limitations and Potential Societal Impacts

The $\ell^\infty$ white-box threat model assumes the availability of the models' gradients, which could present a challenge as such information may not be available to the attacker. It is thus critical to evaluate white-box robustness accurately, as it provides the lower bounds on the robustness of ensemble defenses in practical scenarios.

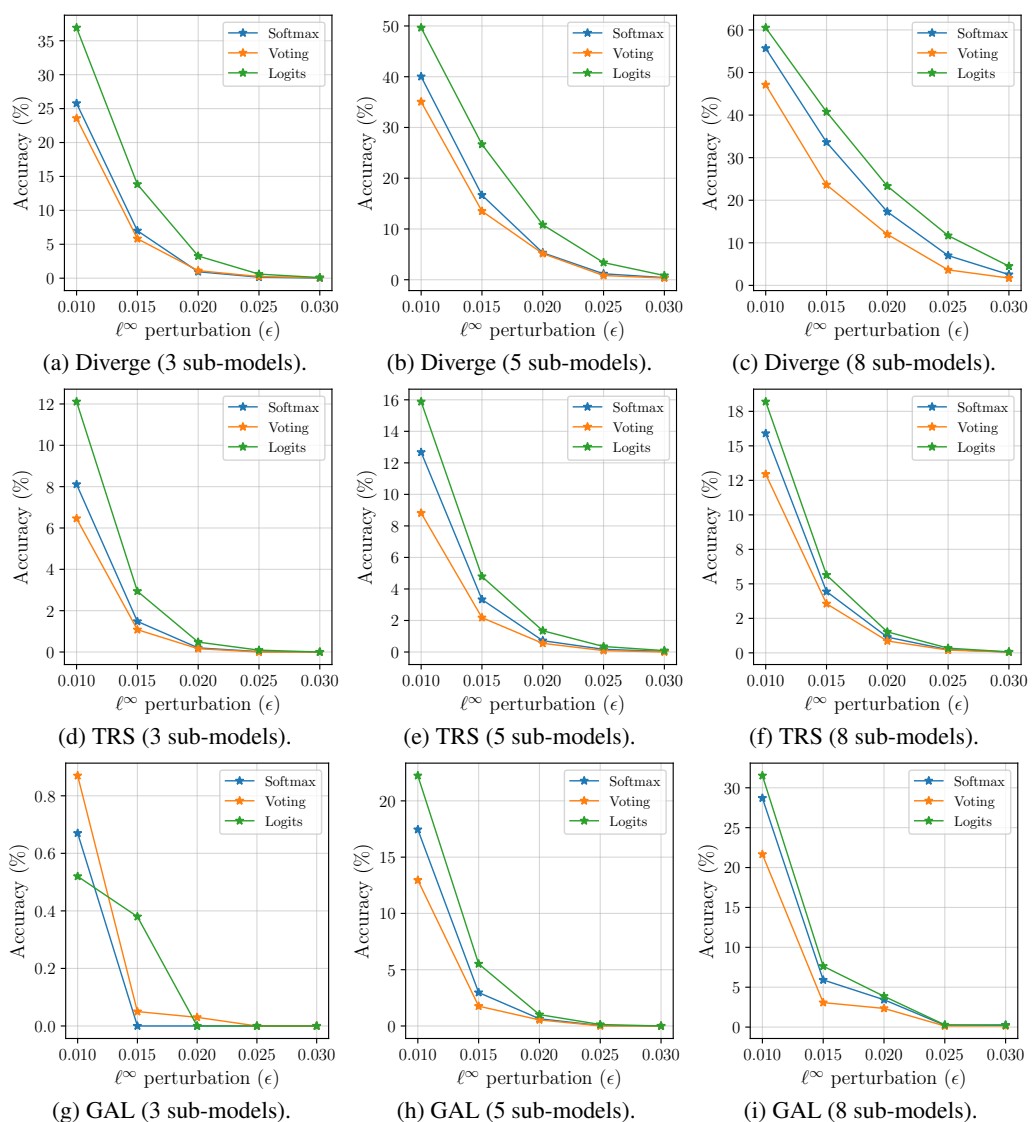

Figure 5: Ensemble defenses with 5 sub-models *w.r.t.* the $\epsilon$ bound on $\ell^\infty$ perturbation. Contrary to existing literatures that propose "softmax"-based ensembles [14, 30, 31], we generally find "logits" to be the most robust option across the three ensemble-forming strategies. All defenses are evaluated with MORA (500 iterations). Note that the only exceptions to this rule, *i.e.*, GAL ensembles with 3 sub-models, exhibit very low robust accuracies.

We acknowledge that adversarial attacks may have the potential to be used by a malicious party, but we believe defending against such attacks is critically pertinent to the accurate evaluation of robustness. We hope this paper furthers the understanding of ensemble robustness, and accurately evaluating adversarial robustness can help improve future defenses. It is also noteworthy that the white-box threat model has applications in the context of advancements in deep learning, and can improve *e.g.*, transfer learning [28, 6], GAN training [2], interpretability [22], and *etc*.

# C Computational Resources

On NVIDIA Tesla V100 GPUs, MORA with 500 iterations uses up to $1.0$ GPU-hours on each ensemble defense, and MORA[mt] uses up to $2.8$ GPU-hours on the CIFAR-10 test set. The run time depends on the number of sub-models in an ensemble and its attack difficulty (Table 7).

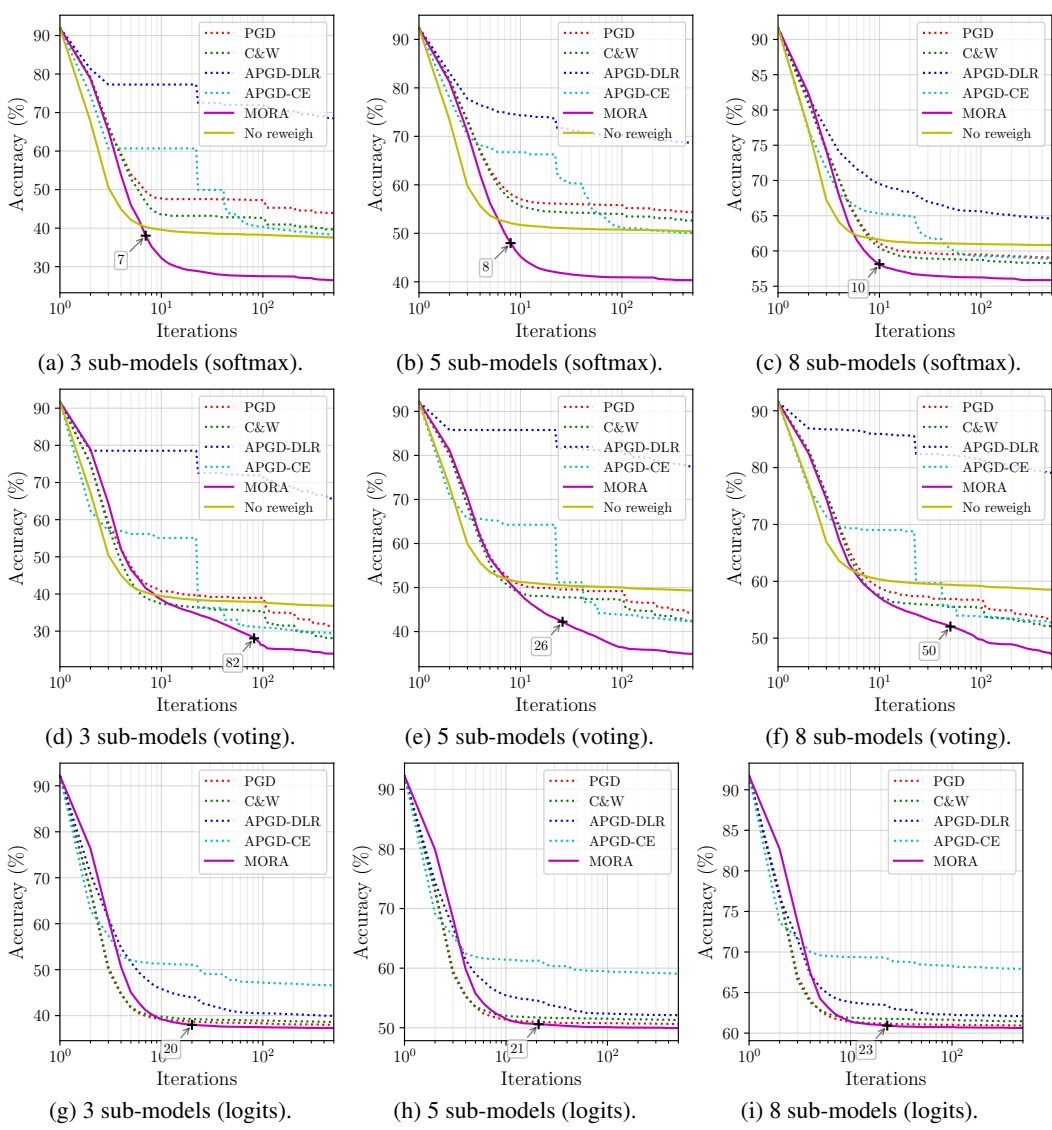

Figure 6: Comparing the convergence speed on Dverge [30] models with three ensemble-forming strategies (softmax, voting, logits) of MORA against a variant of MORA without sub-model reweighing, C&W, PGD and AA losses (APGD-{DLR,CE}). For logit-based ensembles (g, h, i), MORA and "No reweighing" are identical. The horizontal and vertical axes respectively show the iteration count used, and the percentage of unsuccessful images remaining. We annotated the number of iterations for MORA to overtake competition with 500 iterations.

# D   Licenses

Table 8 lists the relevant resources used in this paper and their respective licenses.

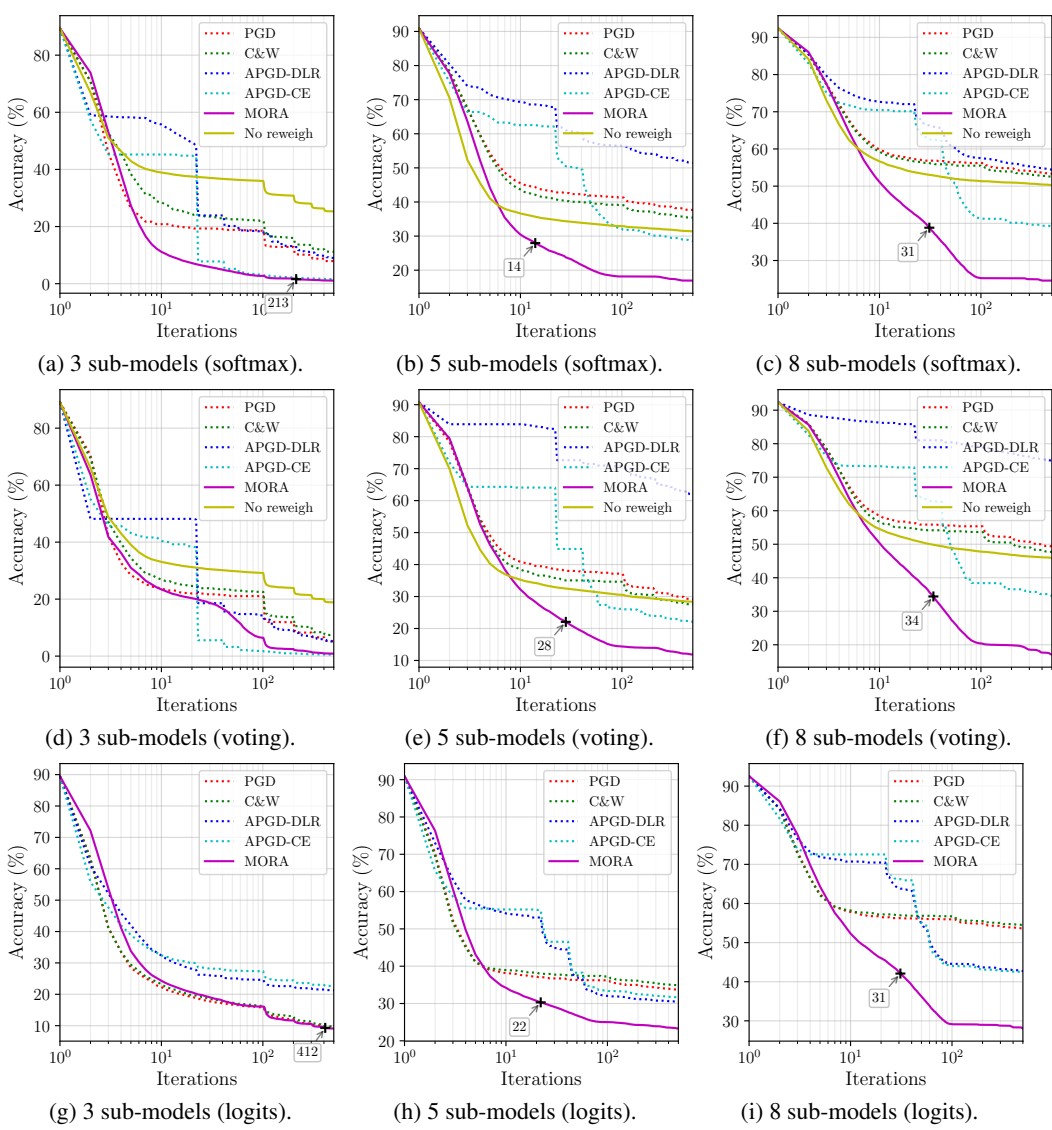

Figure 7: Similar comparisons of convergence rates on GAL [14] models. Please refer to Figure 6 for a detailed description of the setup.

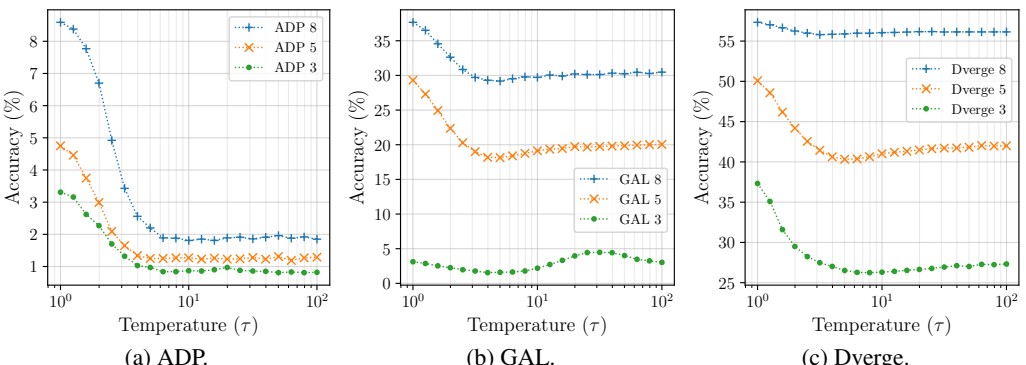

Figure 8: Sensitivity analyses of the temperature coefficient $\tau$ as introduced in (10) and used in $\mathcal{L}_{\beta,\tau}^{\mathrm{mora}}$. All ensembles use softmax values of sub-models to form outputs, as this is used by ADP [19], GAL [14] and Dverge [30]. All defending ensembles share similar flat regions of optimal $\tau$ with low sensitivity.

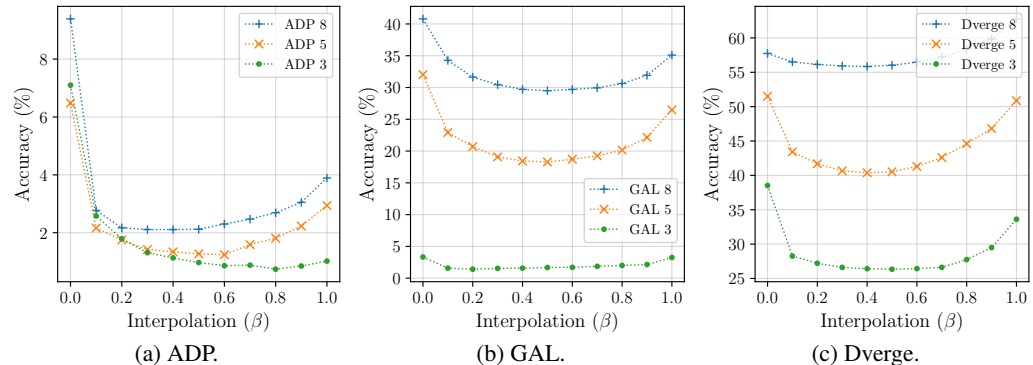

| (a) ADP. | (b) GAL. | (c) Dverge. |

Figure 9: Sensitivity analyses of the $\beta$ interpolation between the original ensemble loss $\mathcal{L}^{\mathrm{sce}}$ and $\mathcal{L}^{\mathrm{mora}}_{\beta,\tau}$, as introduced in (11). All ensembles use softmax values of sub-models to form outputs, as this is used by ADP [19], GAL [14] and Dverge [30]. All defending ensembles share similar flat regions of optimal $\beta$ with low sensitivity.

Table 5: Ablation study of individual components used by MORA. "PGD" is the standard PGD attack [17] with 5 restarts each with 100 iterations. New components are added in consecutive rows. All rows use 500 iterations at most per image, except "Multiple Targets" which additionally uses 100 iterations each for the remaining 9 class labels.

| Defenses | ADP [19] | | | Dverge [30] | | | GAL [14] | | | TRS [31] | | |
|---|---|---|---|---|---|---|---|---|---|---|---|---|
| # | 3 | 5 | 8 | 3 | 5 | 8 | 3 | 5 | 8 | 3 | 5 | 8 |
| **Softmax** | | | | | | | | | | | | |
| PGD | 8.98 | 9.07 | 10.85 | 49.59 | 58.13 | 61.31 | 10.44 | 43.16 | 54.60 | 15.72 | 17.92 | 19.62 |
| + Momentum | 5.98 | 7.10 | 9.22 | 44.49 | 54.61 | 59.13 | 8.13 | 37.59 | 53.39 | 14.01 | 15.91 | 18.02 |
| + Early Stop | 5.04 | 6.57 | 8.89 | 40.18 | 53.53 | 58.84 | 3.70 | 35.34 | 50.53 | 13.21 | 15.63 | 17.97 |
| + Cosine Step Size | 4.14 | 5.08 | 7.65 | 38.09 | 50.71 | 58.36 | 1.34 | 28.60 | 37.17 | 12.26 | 14.88 | 17.32 |
| + Sub-model Logits | 0.69 | 1.13 | 1.84 | 28.51 | 42.91 | 56.95 | 1.78 | 20.73 | 30.51 | 9.15 | 13.52 | 16.80 |
| + Logit Normalization | 0.66 | 0.91 | 1.56 | 27.42 | 41.02 | 55.78 | 2.39 | 19.13 | 29.71 | 9.21 | 13.49 | 16.78 |
| + Adaptive Reweighing | 0.63 | 0.96 | 1.72 | 25.87 | 39.98 | 55.61 | 0.64 | 17.07 | 28.50 | 8.24 | 12.67 | 15.93 |
| + Multiple Targets | **0.34** | **0.67** | **1.32** | **25.26** | **39.50** | **55.57** | **0.51** | **16.05** | **27.44** | **7.60** | **12.47** | **15.64** |
| **Voting** | | | | | | | | | | | | |
| PGD | 13.31 | 13.87 | 13.72 | 36.06 | 49.59 | 57.32 | 7.23 | 34.71 | 51.37 | 13.38 | 15.56 | 17.06 |
| + Momentum | 9.32 | 12.42 | 12.53 | 31.48 | 44.28 | 53.72 | 5.85 | 29.33 | 49.56 | 10.19 | 12.71 | 14.57 |
| + Early Stop | 5.04 | 8.19 | 10.59 | 28.45 | 41.14 | 51.65 | 1.38 | 23.68 | 44.92 | 8.24 | 11.44 | 13.53 |
| + Cosine Step Size | 4.54 | 7.55 | 9.17 | 28.14 | 40.85 | 51.43 | 0.55 | 19.21 | 29.36 | 8.07 | 10.94 | 12.87 |
| + Sub-model Logits | 0.56 | 1.25 | 2.01 | 27.98 | 41.32 | 51.78 | 1.21 | 19.41 | 27.90 | 7.67 | 11.14 | 13.25 |
| + Logit Normalization | 0.73 | 1.15 | 1.81 | 29.79 | 43.87 | 54.30 | 3.82 | 21.08 | 29.27 | 7.68 | 11.13 | 13.29 |
| + Adaptive Reweighing | 0.59 | 1.11 | 2.51 | 23.47 | 34.94 | 47.24 | 0.85 | 12.88 | 21.62 | 5.66 | 8.82 | 11.36 |
| + Multiple Targets | **0.29** | **0.62** | **1.65** | **22.91** | **34.46** | **46.10** | **0.35** | **12.25** | **20.16** | **5.44** | **8.38** | **10.69** |
| **Logits** | | | | | | | | | | | | |
| PGD | 1.55 | 3.29 | 5.32 | 39.20 | 51.28 | 61.22 | 4.39 | 38.27 | 54.64 | 14.17 | 17.78 | 20.39 |
| + Momentum | 0.87 | 1.97 | 3.57 | 39.20 | 50.57 | 60.95 | 10.01 | 33.97 | 53.67 | 13.06 | 16.65 | 19.20 |
| + Early Stop | 0.82 | 1.77 | 3.27 | 37.91 | 50.53 | 60.89 | 8.12 | 31.24 | 50.68 | 12.96 | 16.51 | 19.13 |
| + Cosine Step Size | 0.48 | 1.19 | 2.19 | 37.22 | 49.93 | 60.66 | 0.85 | 22.80 | 31.37 | 12.36 | 16.08 | 18.44 |
| + Sub-model Logits | 0.52 | 1.21 | 2.24 | 37.22 | 49.95 | 60.65 | 0.81 | 22.79 | 31.25 | 12.36 | 16.05 | 18.48 |
| + Logit Normalization | 0.45 | 1.15 | 2.12 | 36.90 | 49.63 | 60.53 | 0.35 | 22.21 | 31.28 | 12.35 | 16.09 | 18.48 |
| + Adaptive Reweighing | 0.47 | 1.14 | 2.07 | 36.88 | 49.65 | 60.52 | 0.52 | 22.12 | 31.57 | 12.10 | 15.85 | 18.22 |
| + Multiple Targets | **0.21** | **0.89** | **1.93** | **36.84** | **49.59** | **60.49** | **0.03** | **19.40** | **30.66** | **12.07** | **15.82** | **18.17** |

Table 6: Comparing the accuracies of SOTA attacks and MORA on PDD+DEG [13] defenses trained with CIFAR-100. Please refer to Table 1 for a detailed explanation. This table uses $\epsilon = 0.01$ as the $\ell^\infty$ perturbation bound, and evaluates its results on the CIFAR-100 test dataset.

| PDD+DEG [13] | # | Clean 1 | Nominal — | PGD 500 | CW 500 | MORA 500 | $A^3$ 12k | AA 4.9k | CAA 1.8k | MORA^mt 1.4k | $\Delta$ |
|---|---|---|---|---|---|---|---|---|---|---|---|
| Softmax | 3 | 79.30 | 22.77 | 11.32 | 14.19 | 1.33 | 9.44 | 9.19 | 11.46 | **0.62** | 22.15 |
| Voting | 3 | 79.28 | 6.55[†] | 2.65 | 2.38 | 0.91 | 0.66 | 1.16 | 1.29 | **0.55** | 6.00 |
| Logits | 3 | 78.94 | 7.25[†] | 3.29 | 1.30 | 0.90 | 0.63 | 0.63 | 0.67 | **0.56** | 6.69 |

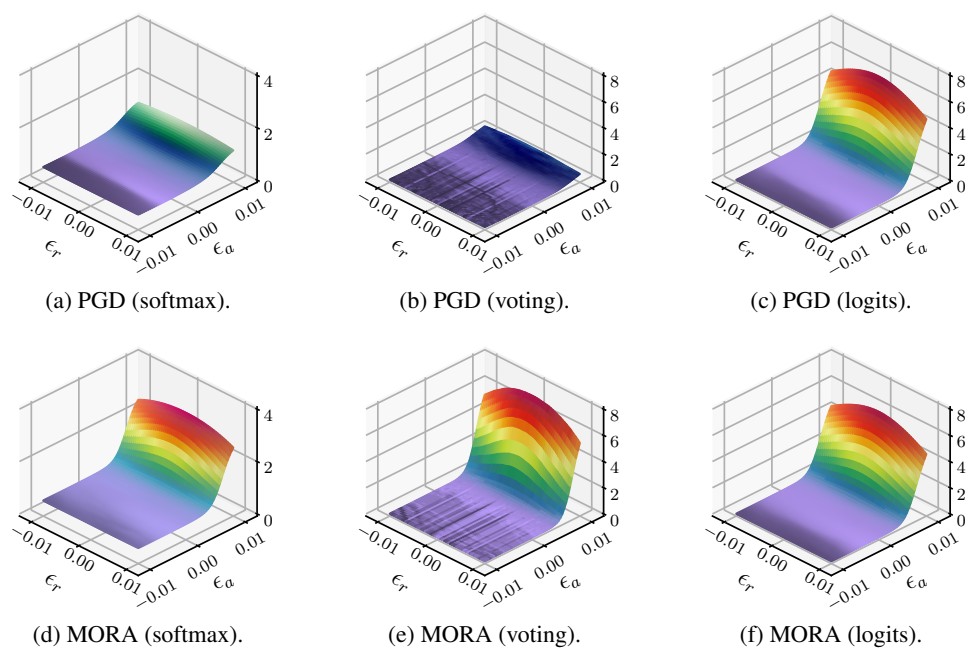

|                   |                   |                   |
|:-----------------:|:-----------------:|:-----------------:|
| (a) PGD (softmax). | (b) PGD (voting). | (c) PGD (logits). |
| (d) MORA (softmax). | (e) MORA (voting). | (f) MORA (logits). |

Figure 10: The averaged loss surfaces across all samples that resisted PGD-10 attacks under all ensemble-forming modes of a 3 sub-model ensemble trained with ADP [19] in the image space $\mathbf{x} + \boldsymbol{g}\epsilon_{\mathrm{a}} + \boldsymbol{g}^{\perp}\epsilon_{\mathrm{r}}$. Here, $\boldsymbol{g}$ denotes the normalized adversarial direction after accumulating 10 initial iterations of gradient updates at the natural input $\mathbf{x}$, and $\boldsymbol{g}^{\perp}$ is its uniformly randomized orthogonal. The top row uses standard PGD attacks, and the bottom row then replaces the SCE loss used in the top row with the MORA loss and uses $\beta = 0.5$.

Table 7: Run time of MORA (up to 500 iterations) and MORA$^{\mathrm{mt}}$ (up to 1.4k iterations) for ADP, Dverge and GAL ensemble defenses on the CIFAR-10 test set. Easier ensembles consume less time with early stopping.

| **Run time** (min) | # | MORA | MORA$^{\mathrm{mt}}$ |
|:---:|:---:|:---:|:---:|
|          | 3 | 2.0 | 3.7 |
| ADP [19] | 5 | 3.4 | 5.8 |
|          | 8 | 6.5 | 11.4 |
|          | 3 | 12.3 | 31.1 |
| Dverge [30] | 5 | 29.7 | 71.2 |
|          | 8 | 61.9 | 169.3 |
|          | 3 | 2.6 | 4.3 |
| GAL [14] | 5 | 15.4 | 35.2 |
|          | 8 | 37.6 | 91.8 |

Table 8: Open-source resources used in this paper.

| Name | License | URL |
|:---:|:---:|:---|
| PyTorch | BSD | GitHub: pytorch/pytorch |
| Dverge | — | GitHub: zjysteven/DVERGE |
| TRS | — | GitHub: AI-secure/Transferability-Reduced-Smooth-Ensemble |
| CIFAR-10 | — | https://www.cs.toronto.edu/~kriz/cifar.html |
| CIFAR-100 | — | https://www.cs.toronto.edu/~kriz/cifar.html |