# OpenReview forum: "MORA: Improving Ensemble Robustness Evaluation with Model Reweighing Attack"
_NeurIPS.cc/2022/Conference — NeurIPS 2022 Accept_

### Official Review · Reviewer_8pbN · 2022-07-09

**Rating:** 5
**Confidence:** 5
**Soundness:** 3 good
**Presentation:** 3 good
**Contribution:** 2 fair

**Summary:**

In this paper, the authors propose an adversarial attack (MORA) against model-ensemble-based defense by reweighing the importance of sub-models. The authors first investigate the factors which cause existing white-box attacks to fail against several state-of-art model-ensemble defenses, that is gradient obfuscation. Then, the authors propose a strategy for crafting adversarial samples by conducting optimization on several 'important' sub-models within ensembled models.  Motivated by such a strategy, the authors propose a MORA loss, which incorporates a sum of the weighted variant of sub-model logits, in order to
expose sub-model logits with adaptive reweighing. Through extensive empirical evaluation, MORA is proved to be able to bypass existing SOTA model ensemble defense approaches. The authors further compare MORA with several state-of-art white-box attacks(PGD, C&W attacks) in both the dimensions of efficiency(convergence rate) and attack efficacy (attack success rate).  The results show that MORA can significantly outperform existing white-box attacks against model ensemble defense.

**Questions:**

I have several questions:

I. What's the motivation for developing a white-box attack against SOTA defense methods against black-box attacks?

II. Can you compare MORA with several decision-based attacks?



**Limitations:**

See above.

**Strengths And Weaknesses:**

=====Strengths=====
I. This work is well-structured and easy to follow.

II. This work proposes an effective method MORA and solves an interesting problem.

III. The evaluation is comprehensive and convincing.


=====Weakness=====

I have two main concerns about this work.

I. Limited Novelty and Practicality. This work seems to be a kind of 'defense-aware attacks' approach which built upon the existing defense approach(i.e., model-ensemble defense). The proposed attack can only bypass a specific category of defense, which limits its practicality. Moreover, the proposed attack is evaluated under the white-box settings, however, as mentioned in the abstract and introduction, the ensemble-based defense approach is mainly used against black-box attacks (i.e., transferable attacks).  In general, there's no evidence to show that ensemble-based defense is or performs close to the state-of-art defense against white-box adversarial attacks. Therefore, the motivation for developing a white-box attack bypassing ensemble-based defenses that are mainly used against black-box attacks is unclear to me.

II. Evaluation. Indeed, I am convinced that MORA can outperform existing white-box attacks against ensembled-based defense through empirical evaluation. However, I think selecting current white-box attacks for comparison is not enough. This is because existing white-box attacks are prone to gradient obfuscations, which is also claimed in your introduction section and other related work.  In contrast, I think several decision-based attacks (hop skip jump attack, Sign-Flip attack.) is more suitable for this scenario.  Even though these decision-based attacks are built under black-box, they can also be applied in the white-box settings. I have expertise in such decision-based attacks and observe that decision-based attacks can bypass gradient-obfuscation defense. I think you should also compare MORA with them.

---

> ### Author Response · Authors · 2022-08-02
> **Thank you for your detailed comments, we would like to address your concerns below.**
>
> Thank you for providing a detailed feedback on the paper, and we would like to address your concerns below.
>
> 1. We would like to answer your concerns on novelty and practicality separately:
>     1. Please note that the proposed MORA attack is not limited to ensemble defenses. Other defense methods can be considered as degenerate ensemble defenses with only 1 sub-model. [In a separate comment](https://openreview.net/forum?id=d_m7OKOmPiM&noteId=tgG7VacNe1y), we provide a table of comparisons against other SOTA attacks on such defenses, and show that MORA can be competitive against recent SOTA attacks.
>     2. We would like to argue that the adversarial robustness of ensemble defenses in the white-box threat model is an interesting topic, and this paper brings a notable amount of new discoveries. First, besides adversarial training, it presents a new opportunity to improve model robustness while preserving clean accuracy. Second, existing strong attacks on other defenses perform surprisingly poorly on ensembles, and new defenses should go beyond existing evaluation baselines. The sizeable robustness overestimation would thus present a significant obstacle to improve the SOTA on ensemble defenses. Third, it also provides empirical lower bounds on gray- and black-box defenses. Finally, we are first to show that it is not trivial to combine the benefits of ensemble defenses with adversarial training, as doing so often does not improve robustness.
>
> 2. To address your concerns regarding the lack of decision-based attacks that may potentially work around the challenge of gradient obfucation, we would like to present our findings in the table below as an extension to Table 1 in the paper. Note that both [Square attack [1]](https://arxiv.org/abs/1912.00049) and HopSkipJump (HSJ) attack use 5000 queries per image, while the standard PGD uses 100 iterations with 5 random restarts. For Sign-flip attack, an initial result using 5k queries on ADP with 3 sub-models still noticeably overestimates its robustness (43.14\%). It is noteworthy that none of the decision-based attacks can outperform the PGD-500 baseline reported in Table 1 in the paper.
>
> | **Softmax**  | Clean | PGD   | Square | HSJ   |   | **Voting**   | Clean | PGD   | Square | HSJ   |   | **Voting**   | Clean | PGD   | Square | HSJ   |
> |--------------|-------|-------|--------|-------|---|--------------|-------|-------|--------|-------|---|--------------|-------|-------|--------|-------|
> | **ADP 3**    | 92.88 | 5.98  | 46.52  | 10.93 |   | **ADP 3**    | 91.84 | 9.32  | 83.08  | 10.59 |   | **ADP 3**    | 92.86 | 0.87  | 34.88  | 11.16 |
> | **ADP 5**    | 93.34 | 7.10  | 48.09  | 15.44 |   | **ADP 5**    | 93.13 | 12.42 | 82.62  | 14.81 |   | **ADP 5**    | 93.48 | 1.97  | 40.35  | 18.05 |
> | **ADP 8**    | 93.48 | 9.22  | 51.28  | 20.05 |   | **ADP 8**    | 93.28 | 12.53 | 82.40  | 28.60 |   | **ADP 8**    | 93.38 | 3.57  | 45.56  | 23.42 |
> | **GAL 3**    | 89.41 | 8.13  | 58.90  | 21.52 |   | **GAL 3**    | 89.09 | 5.85  | 84.53  | 28.70 |   | **GAL 3**    | 89.50 | 10.01 | 62.67  | 27.30 |
> | **GAL 5**    | 90.93 | 37.59 | 63.34  | 45.94 |   | **GAL 5**    | 90.77 | 29.33 | 85.46  | 55.75 |   | **GAL 5**    | 90.93 | 33.97 | 65.18  | 58.92 |
> | **GAL 8**    | 92.45 | 53.39 | 66.26  | 63.15 |   | **GAL 8**    | 92.37 | 49.56 | 86.68  | 71.29 |   | **GAL 8**    | 92.54 | 53.67 | 66.61  | 65.60 |
> | **Dverge 3** | 91.99 | 44.49 | 67.38  | 54.38 |   | **Dverge 3** | 91.72 | 31.48 | 88.22  | 53.29 |   | **Dverge 3** | 92.19 | 37.99 | 68.96  | 56.63 |
> | **Dverge 5** | 92.38 | 54.61 | 72.15  | 65.34 |   | **Dverge 5** | 92.18 | 44.28 | 88.41  | 69.33 |   | **Dverge 5** | 92.28 | 50.57 | 73.88  | 69.54 |
> | **Dverge 8** | 91.65 | 59.13 | 75.46  | 71.98 |   | **Dverge 8** | 91.58 | 53.72 | 87.03  | 75.01 |   | **Dverge 8** | 91.73 | 60.95 | 76.51  | 73.61 |
>
> We appreciate your review and we hope the above answers could address your concerns adequately. Finally, we would like to kindly ask the reviewer to reconsider the current rating and its criteria, as the review is overall positive in terms of MORA's effectiveness, the challenge being interesting, notes that the "evaluation is comprehensive and convincing". and we also strive to ensure the results are reproducible with the source code.
>
> [1]: Andriushchenko et al., Square Attack: a query-efficient black-box adversarial attack via random search, ECCV 2020.

---

> > ### Comment · Reviewer_8pbN · 2022-08-08
> > **Response to the rebbutals**
> >
> > Thanks for your rebuttals. It addresses my concerns. Therefore, I raise my score to 5 accordingly.

---

> ### Author Response · Authors · 2022-08-02
> **Comparing attacks on standard non-ensemble defenses**
>
> In the following table, we reproduce the attacks on CIFAR-10 with [FAB](https://proceedings.mlr.press/v119/croce20a.html), AAA, AA, CAA and MORA on popular defenses that demonstrate high robustness. For these defending models, we consider each model to be a degenerate ensemble with 1 sub-model. To introduce variability between the final logit and the MORA branches (as in Fig. 2), the logit normalization in the MORA branch instead uses $1[\mathbf{z}\_y - \mathbf{z}\_{\hat{y}} > 0] \cdot z / \mathrm{detach}(\mathrm{relu6}(\mathbf{z}\_y - \mathbf{z}\_{\hat{y}}))$.
>
> | Defense                  | Clean | Nominal | PGD 100 | FAB 100 | AAA 5k | AA 4.9k | CAA 1.8k | MORA 1.4k |
> |--------------------------|-------|---------|---------|---------|--------|---------|----------|-----------|
> | [Wang et al., Misclassification-aware (ICLR 2020)](https://openreview.net/forum?id=rklOg6EFwS)  | 87.50 | 65.04   | 62.55   | 57.60   | 56.27  | 56.29   | 56.30    | **56.21**     |
> | [Hendrycks et al., Pre-training (ICML 2019)](https://arxiv.org/abs/1901.09960)             | 87.11 | 57.40   | 57.54   | 55.58   | 54.89  | 54.92   | 54.87    | **54.78**     |
> | [Wong et al., Fast adversarial training (ICLR 2020)](https://openreview.net/forum?id=BJx040EFvH) | 83.34 | 46.06   | 46.52   | 44.25   | 43.44  | 43.41   | 43.43    | **43.37**     |
> | [Zhang et al., TRADES (ICML 2019)](https://proceedings.mlr.press/v97/zhang19p.html)                   | 84.92 | 56.43   | 55.50   | 53.96   | 53.03  | 53.08   | 53.06    | **53.00**     |

---

### Official Review · Reviewer_qYQF · 2022-07-11

**Rating:** 6
**Confidence:** 4
**Soundness:** 3 good
**Presentation:** 4 excellent
**Contribution:** 2 fair

**Summary:**

This paper introduces MORA, a novel adversarial attack against ensembles. The attack is motivated by the fact that existing white-box attacks over estimate the robustness of ensembles, either due to gradient obfuscation or the difficulty of simultaneously fooling diverse gradients. MORA is designed to adapt its focus to the models in the ensemble. Experiments are performed on CIFAR-10 and -100 with ResNet20 in white-box setting. MORA is compared against other attacks and multiple ensemble defenses.

**Questions:**

- In Alg. 1, it seems the adversarial sample obtained after a maximum of $I$ iterations is used. What are the odds that it is the strongest sample? Should the best sample out of the $I$ iterations not be used instead (see also [1])?
- Tab. 2 presents results of adversarially trained models ensembled using Dverge defense. What happens when attacking with MORA an ensemble of (individually) adversarially trained models without ensemble defenses?
- Does it makes sense to compare attacks convergence in terms of number of iterations (Fig. 4)? Do they perform the same computations in each iterations? Does it also make sense to compare convergence over computation time?
- Are the results presented for CW attack also under $L_\infty$ norm?

References:
- [1] Pintor M. et al. Indicators of Attack Failure: Debugging and Improving Optimization of Adversarial Examples. https://arxiv.org/abs/2106.09947

**Limitations:**

The limitations of the threat model are addressed in the appendix.

**Strengths And Weaknesses:**

Strengths:
- The design of the loss function used for MORA seems sound and well-motivated by the causes of attack failures on ensembles.
MORA incorporates state-of-the-art community practices for thorough robust evaluation.
- Under small perturbation budgets, MORA seems to find adversarial examples that other attacks do not, thus providing a better estimate of robustness for the given ensembles.
- The experiments are extensive, comparing the proposed attack against state-of-the-art white-box attacks and ensemble defenses, under different aggregation rules and ensemble sizes. An ablation analysis is also provided.
- Very well written paper.

Weaknesses:
- MORA seems to lose its advantage over other attacks under certain regimes. Results are almost on par with A$^3$ and the nominal values in Tab. 2.
- It is standard to use perturbation budget $\epsilon = 8 / 255$ on CIFAR-10. However, the main results table in the paper (Tab. 1) uses $\epsilon=0.01$, just under a third of the standard value. Results for $\epsilon=0.03$ are included in the appendix, where we see that the gap in ensemble robustness estimation significantly narrows between general-purpose white-box attacks and the proposed MORA. The numbers are on par for most rows, diminishing the claims of the paper.
- Despite the extensive experiments, some aspects of the evaluation are limited. Particularly, focusing only on ResNet20 architecture, CIFAR data and $L_{\infty}$ robustness does not convince beyond a doubt that MORA is the new adversarial standard for ensemble evaluation.

Minor:
- It would be a plus to start by stating the threat model. It become apparent later in the paper that the assumption is white-box access althroughout.
- L135: there exists -> there exist
- L186: Base on this property -> based on this property
- L216: attackss -> attacks

---

> ### Author Response · Authors · 2022-08-02
> **Thanks for your positive comments, your concerns addressed below.**
>
> Thank you for providing a detailed review on the paper.  We appreciate your positive feedback, and would like to address your concerns in this section below.
>
> 1. We would like to highlight that $\mathrm{A}^3$ is the most compute-intensive SOTA, which uses $8.5\times$ more computational resources than MORA --- it requires 12k iterations to obtain the results, whereas the multi-targeted MORA uses only 1.4k iterations. Given identical budget, MORA is expected to be stronger than the competing baselines.
>
> 2. The reasons for the low $\epsilon$ perturbations is because none of the ensembles are adversarially trained, except for the ones in Table 2. Under $\epsilon = 0.03$, most models have close to zero robustness under strong attacks, so we can expect the gap to diminish. However, we kindly point out that MORA is more computational efficient than the other competing methods that comprise a large arsenal of attack algorithms, uses only a unified algorithm, while being highly effective.
>
> 3. Please kindly note that we also included a PDD+DEG, an ensemble defense on CIFAR-100. We are happy to include more results, but we are limited by the number of open-source defenses in existing literatures. To this end, we thus make MORA open source and maintain a leaderboard that encourages others to submit their defenses.
>
> Thanks for pointing out the typos in the paper, and we would like to answer your remaining questions below:
>
> 1. We kindly point out that line 8 of Alg. 1 performs an early exit upon successful attack. The algorithm is thus not looking for the strongest adversarial examples (although it can do so by removing the early exit), but rather it tries to improve the computational efficiency of the attack.
>
> 2. We are working on this result and would like to get back to you in the next few days.
>
> 3. All attack methods examined in the paper perform gradient steps on the image, the differences between them exist in the image parameter updates, which consumes only 3x32x32x7 = 21.5k multiply-accumulate operations (MACs) per image and the loss function uses only a few hundred MACs in forward and backward passes, whereas an ensemble model with 8 sub-models consumes 2x331M MACs per image for both forward and backward passes. It is notable that the majority of the computational cost (> 99.99\%) is thus spent on model forward and backward passes.
>
> 4. Yes, the C\&W results in the paper are also under the $\ell_\infty$ norm to ensure identical constraints.

---

> > ### Author Response · Authors · 2022-08-08
> > **Additional updates to our response.**
> >
> > To further address the first point of concerns, for Table 2 we add an additional column entry of MORA 10k, i.e. increasing the number of maximum iterations to 10k. Specifically, for the 1 untargeted and 9 targeted variants, we vary $\beta \in \\{0, 0.25, 0.5, 0.75, 1\\}$, and use 200 iterations for each configuration ((1 + 9) * 5 * 200 = 10 000), giving a total of 10k iterations. With increased computational effort, MORA is competitive with AAA with 12k iterations.
> >
> > | **Softmax**  | Clean | AAA 12k | MORA 10k  | **Voting**   | Clean | AAA 12k   | MORA 10k  | **Logits**   | Clean | AAA 12k | MORA 10k  |
> > |--------------|-------|---------|-----------|--------------|-------|-----------|-----------|--------------|-------|---------|-----------|
> > | **Dverge 3** | 83.78 | 42.66   | **42.62** | **Dverge 3** | 83.74 | 42.66     | **42.60** | **Dverge 3** | 83.67 | 38.78   | **38.24** |
> > | **Dverge 5** | 86.09 | 40.74   | **40.71** | **Dverge 5** | 86.03 | 40.91     | **40.82** | **Dverge 5** | 86.05 | 36.42   | **36.01** |
> > | **Dverge 8** | 86.69 | 39.33   | **39.29** | **Dverge 8** | 86.65 | **39.43** | **39.43** | **Dverge 8** | 86.54 | 34.89   | **34.03** |
> >
> > Regarding the second question, *"what happens when attacking with MORA an ensemble of (individually) adversarially trained models without ensemble defenses?"* We performed this experiment exactly and we find them to be more robust than adversarially trained Dverge ensembles from Table 2. Here "AT $n$" denotes ensembles with $n$ adversarially trained sub-models. Larger ensembles are most robust when forming ensembles with the "logits" strategy, and "voting" shows decreasing robustness with larger ensembles. Combining both advantages of ensemble-based defenses and adversarial training is thus notably not trivial.
> >
> > | **Softmax**  | Clean | PGD 500 | C\&W 500 | MORA 1.4k | **Voting**   | Clean | PGD 500 | C\&W 500 | MORA 1.4k | **Logits**   | Clean | PGD 500 | C\&W 500 | MORA 1.4k |
> > |--------------|-------|---------|----------|-----------|--------------|-------|---------|----------|-----------|--------------|-------|---------|----------|-----------|
> > | **AT 3** | 78.62 |  48.27  |   46.97  |   45.30   | **AT 3** | 78.36 |  60.84  |   60.61  |   40.17   | **AT 3** | 78.56 |  48.19  |   47.06  |   45.37   |
> > | **AT 5** | 80.12 |  49.39  |   47.84  |   46.51   | **AT 5** | 79.58 |  55.09  |   55.22  |   40.31   | **AT 5** | 80.05 |  49.22  |   47.96  |   46.50   |
> > | **AT 8** | 80.86 |  49.92  |   48.49  |   46.82   | **AT 8** | 80.60 |  49.82  |   49.57  |   39.41   | **AT 8** | 80.81 |  49.48  |   48.66  |   46.90   |

---

> > > ### Comment · Reviewer_qYQF · 2022-08-08
> > > **Thank you for your response**
> > >
> > > Thank you for the detailed response and additional experimental results. I am sure these will strengthen the paper.

---

### Official Review · Reviewer_9Ruy · 2022-07-11

**Rating:** 5
**Confidence:** 4
**Soundness:** 2 fair
**Presentation:** 3 good
**Contribution:** 2 fair

**Summary:**

This paper proposes an attack targetting the model ensemble defenses. The main idea is to assign different weights to different models. The values of the weights depend on the models' contribution to outputs. The results demonstrate that the proposed attack outperforms existing ones.

**Questions:**

1. Can MORA be extended to black-box attack scenarios or be used to attack other defenses? The application scenario is limited. It would be more exciting to see some extensions.

2. There are some typos in the existing version. For example, should it be 'A successful attack happens when $k^{[E]} < 0$'?

**Limitations:**

Yes

**Strengths And Weaknesses:**

Strengths: The idea is simple and easy to understand. The experiments show

Weaknesses: Overall, the contributions are incremental and the application scenario restricts to white-box ensemble defenses.

---

> ### Author Response · Authors · 2022-08-02
> **Thanks for your comments, concerns addressed below.**
>
> We would like to thank the reviewer for the constructive feedback, and would like to answer your questions below.
>
> To address your concern, we would like to argue that the adversarial robustness of ensemble defenses in the white-box threat model is an interesting topic. First, besides adversarial training, it presents a new opportunity to improve model robustness while preserving clean accuracy. Second, existing strong attacks on other defenses perform surprisingly poorly on ensembles. The sizeable robustness overestimation would thus present a significant obstacle to improve the SOTA on ensemble defenses. Finally, it also provides an empirical lower bound on gray- and black-box defenses.
>
> To answer your questions:
> 1. Thanks for the kind suggestion, we are also looking to include evaluation of black-box transferred attacks in the future. We also would like to highlight that the proposed MORA attack strategy is not limited to ensemble defenses, as standard single model defenses are special cases of ensemble defenses (single sub-model). [In a separate comment](https://openreview.net/forum?id=d_m7OKOmPiM&noteId=tgG7VacNe1y), we provide a table of comparisons against other SOTA attacks on such defenses.
>
> 2. Thanks for pointing out the typos, and we have updated the paper accordingly.

---

### Official Review · Reviewer_GcPa · 2022-07-12

**Rating:** 6
**Confidence:** 4
**Soundness:** 3 good
**Presentation:** 3 good
**Contribution:** 3 good

**Summary:**

This paper focuses on the problem of robustness evaluation for ensemble-based defense methods. Authors argue that existing SOTA white-box attack methods may overestimate the robustness of ensemble defenses because of two reasons, i.e., (1) the ensemble-forming strategy used in ensemble defenses could lead to gradient obfuscation and (2) attacking would fool majority sub-models, while the ensemble model's output may still correct. In order to provide a more effective and reliable way to evaluate ensemble defenses, this paper proposes a novel attack method that adaptively assigns different weights for different sub-models based on their contributions to the attack gradient. To evaluate the effectiveness of the proposed attack method, authors compare it's attack performance with several SOTA white-box attack methods on four ensemble defenses under three ensemble-forming strategies. Experimental results verify that the proposed method outperforms baseline methods with large margin and converges much faster.

**Questions:**

1. Is it possible to verify the two reasons that cause the existing SOTA white-box attack methods overestimate the robustness of ensemble defenses through either empirical study or theoretical analysis? Although the two reasons authors pointed out in this paper sound reasonable, it would be better if more convincing evidence can be provided to further verify these two reasons.

2. Some observations from the experimental results are very interesting. (1) As shown in Table 2, larger ensembles lead to reduced robustness when sub-models are adversarially trained facing with any attack methods listed in the table. However, ideally the adverasrially trained sub-models could be more robust against attacks and, hence, the larger ensemble of robust sub-models should provide improved robustness performance, comparing with small number of sub-models. Can authors provide some understanding about this observation? Does this observation reveal some possible drawbacks/limitations of applying the ensemble idea to improve model's robustness? (2) Since the majority votes ensemble strategy requires attacks to break at least half of sub-models, intuitively the robustness of ensemble defense methods based on the majority votes strategy should be higher than other two ensemble strategies. However, the experimental results give an opposite observation. Can authors give some discussion about this observation? Does this observation caused by the simplicity of the majority votes strategy (e.g. comparing with other two strategies, the majority votes only focuses on the final outputs and ignore some useful information such as logits)?

3. Some notations are not clear. (1) Does the three star signs with pick color in Figure 1 and 2 represent the attack performance of the MORA method on some specific ensemble defense method under three different ensemble-forming strategies? If so, it would be better if each star can be denoted more specifically. (2) Does $\text{MORA}^{mt}$ represent the multi-targeted version of MORA? If so, it would be better if some introduction about this variant of MORA can be provided in the beginning of the experiment part. (3) What's the meaning of "Nominal" appeared in both Figure 1&2 and Table 1&2? Is it the robust accuracy reported in corresponding original papers?


**Limitations:**

The limitations and potential societal impacts are discussed in the appendix of this paper. As pointed by the authors, although the attack method proposed in this paper may be used by a malicious party, an effective attack method can boost the development of robust DNN models in the future.

**Strengths And Weaknesses:**

Strengths:

1. This paper investigates the robustness of ensemble defenses under multiple ensemble-forming strategies and provides extensive empirical results.
2. The proposed method achieves much improved success rate on ensemble defense methods and convergence rate than SOTA attack methods among different experimental settings, which sufficiently demonstrate the superiority of the proposed method.
3. Authors make the source code available and also maintain a leaderboard of ensemble defenses under various attack strategies, which can boost the research on this direction.


Weaknesses:

Although experimental results verify the effectiveness of the proposed attack method, the intuition/motivation of the designing of this method is not quite clear. Specifically, it's not clear why the sub-model reweighing technique is able to produce a stronger attack method than SOTA white-box attack methods for ensemble defenses. For more detailed weaknesses, see the following questions.

---

> ### Author Response · Authors · 2022-08-02
> **Thanks for the detailed comments, your concerns addressed below.**
>
> We would like to thank the reviewer for providing a detailed feedback on our paper. We appreciate the effort that went into the review, and would like to address your concerns below.
>
> 1. To verify the cause of gradient obfuscation, we additionally introduced Fig. 10 (page 20) in the appendix (A.5, line 479), which visualizes the loss surfaces of an ADP defense with the three ensemble-forming strategies under PGD and MORA attacks. Flat loss surface indicate the presence of gradient obfuscation, as the accumulated adversarial gradient direction is less effective in increasing the loss. We provide discussions for the two causes below:
> 	1. *Gradient obfuscation via ensemble-forming strategies*: Under PGD-10 attacks, both softmax- and voting based ensembles (Fig. 10a and 10b) exhibit gradient obfuscation to some extent when compared with the logic-based variant (Fig. 10c).
> 	2. *Gradient diversification*: As sub-model gradients counteract, PGD attacks on softmax- and voting-based ensembles may result in an averaged gradient direction $\mathbf{g}$ that experience difficulty in increasing loss (Fig. 10a and 10b respectively). Adopting sub-model reweighing (bottom row in Fig. 10) alleviates this difficulty, and allows the attack to succeed more reliably.
> 2. We thank the reviewer for raising insightful questions regarding our observations, and would like to response to your questions:
> 	1. *Lower robustness with larger adversarially trained ensembles*: To answer this, we perform additional experiments on adversarially trained ensembles, and discover that as we increase the number of sub-models, the robustness of individual sub-models decreases, as shown in the tables below. Here, each table corresponds to a softmax-based Dverge model with different sub-model counts, and we report the robustness of each sub-model using MORA to attack a degenerate case of an ensemble of 1 sub-model. This may suggest that the sub-models may not have sufficient capacity to adversarial training while learning to stop sub-model transferability jointly, and more sub-models increases the number of learning objectives. This could thus hurt the performance of the resulting ensemble.
>
>     | Dverge 3   | 1     | 2     | 3     |
>     |------------|-------|-------|-------|
>     | Clean      | 77.29 | 76.47 | 77.60 |
>     | Robustness | 36.36 | 35.94 | 36.38 |
>
>     | Dverge 5   | 1     | 2     | 3     | 4     | 5     |
>     |------------|-------|-------|-------|-------|-------|
>     | Clean      | 76.43 | 78.33 | 76.55 | 78.39 | 78.15 |
>     | Robustness | 30.31 | 31.91 | 31.25 | 32.39 | 31.33 |
>
>     | Dverge 8   | 1     | 2     | 3     | 4     | 5     | 6     | 7     | 8     |
>     |------------|-------|-------|-------|-------|-------|-------|-------|-------|
>     | Clean      | 76.97 | 77.80 | 80.55 | 78.30 | 76.99 | 75.71 | 77.57 | 77.13 |
>     | Robustness | 26.37 | 27.03 | 26.38 | 26.88 | 26.88 | 25.89 | 25.89 | 25.80 |
>
>     2. *Majority voting is less robust when compared to the other two modes*: It is true that when $>50\\%$ of all sub-models give correct predictions, the ensemble must be correct. However, we would like to highlight that this is a relatively infrequent occurrence if the model demonstrate low robustness. For instance, The following table shows the image counts v.s. the numbers of remaining robust sub-models after MORA with 100 iterations, and their respective number of successful attacks.  For ADP with 8 sub-models, only $14.08\\%$ of the CIFAR-10 test images are correctly classified simultaneously by $\geq4$ sub-models. We hypothesize this behavior is image-dependent, and sub-models are most likley either majority fooled, or none fooled.
>
>     | # robust sub-models | 0    | 1    | 2    | 3    | 4   | 5 | 6 | 7 | 8   |
>     |---------------------|------|------|------|------|-----|---|---|---|-----|
>     | # images            | 1420 | 1789 | 2380 | 3003 | 870 | 0 | 0 | 0 | 538 |
>     | # success           | 1420 | 1789 | 2380 | 3003 | 870 | 0 | 0 | 0 | 0   |
>
> 1. Thank you for pointing out the notations that are not clear.
> 	1. The three star signs denote the attack performance of MORA on the defense models reproduced directly from official source codes. We would like to highlight that the three points have different computational costs (100, 500 and 1.4k iterations on the x-axis from left to right respectively), and thus they report different accuracies under attack (y-axis). We can certainly include additional figures for the other defenses and ensemble-forming modes.
> 	2. In lines 245-246 in the paper, we define $\mathrm{MORA}^{\mathrm{mt}}$ as the multi-targeted variant that targets the remaining 9 class labels with 100 iterations each. This fixes $\beta = 0.5$ for all 9 targets and we updated the paper to reflect this.
> 	3. “Nominal” values in Figures 1-2 and Tables 1-2 are either reported in the original paper, or reproduced with official source code. To differentiate the latter, we mark those with $\dagger$.

---

> > ### Comment · Reviewer_GcPa · 2022-08-07
> > **Response to authors' rebuttal**
> >
> > Thanks authors' replies that address most of my concerns. Based on the overall quality of the work, I will keep my original score.

---

### Author Response · Authors · 2022-08-08
**Follow up on our responses.**

Dear Reviewers,

We appreciate your thoughtful reviews. We kindly remind that the discussion period ends on Aug 9 and we are available to answer any questions you may have after our responses.

---

> ### Author Response · Authors · 2022-08-09
> **Thanks for engaging in discussion.**
>
> Please let us know if you have any unresolved concerns, and we would be happy to answer them in the remaining next few hours before discussion period ends to further strengthen our submission.

---

### Meta-Review · Area_Chair_HuGA · 2022-08-27

**Recommendation:** Accept
**Confidence:** Certain

**Metareview:**

This paper introduces a new way to generate adversarial examples for defenses that work by ensembling over many different independent predictors. The reviewers all liked this paper and believe that the results were a useful improvement on prior work. While the margin of improvement is not massive (and it is a smaller gap as perturbations get larger) but the improvement appears correct and is well explained.

**Award:**

No

---

### Decision · Program_Chairs · 2022-09-14

Accept